# How many measurements are enough? Bayesian recovery in inverse problems with general distributions

**Ben Adcock**
Department of Mathematics
Simon Fraser University
Canada

**Nick Huang**
Department of Mathematics
Simon Fraser University
Canada

## Abstract

We study the sample complexity of Bayesian recovery for solving inverse problems with general prior, forward operator and noise distributions. We consider posterior sampling according to an approximate prior $\mathcal{P}$, and establish sufficient conditions for stable and accurate recovery with high probability. Our main result is a non-asymptotic bound that shows that the sample complexity depends on (i) the intrinsic complexity of $\mathcal{P}$, quantified by its *approximate covering number*, and (ii) concentration bounds for the forward operator and noise distributions. As a key application, we specialize to generative priors, where $\mathcal{P}$ is the pushforward of a latent distribution via a Deep Neural Network (DNN). We show that the sample complexity scales log-linearly with the latent dimension $k$, thus establishing the efficacy of DNN-based priors. Generalizing existing results on deterministic (i.e., non-Bayesian) recovery for the important problem of random sampling with an orthogonal matrix $U$, we show how the sample complexity is determined by the *coherence* of $U$ with respect to the support of $\mathcal{P}$. Hence, we establish that coherence plays a fundamental role in Bayesian recovery as well. Overall, our framework unifies and extends prior work, providing rigorous guarantees for the sample complexity of solving Bayesian inverse problems with arbitrary distributions.

## 1 Introduction

Inverse problems are of fundamental importance in science, engineering and industry. In a standard setting, the aim is to recover an unknown vector (e.g., an signal or image) $x^* \in \mathbb{R}^n$ from *measurements*

$$y = Ax^* + e \in \mathbb{R}^m. \tag{1.1}$$

Here $e \in \mathbb{R}^m$ is measurement noise and $A \in \mathbb{R}^{m \times n}$, often termed the *measurement matrix*, represents the forwards operator. While simple, the discrete, linear problem (1.1) is sufficient to model many important applications [5, 25, 64, 68]. It is common to solve (1.1) using a Bayesian approach (see, e.g., [30, 71]), where one assumes that $x^*$ is drawn from some prior distribution $\mathcal{R}$. However, in practice, $\mathcal{R}$ is never known exactly. Especially in modern settings that employ Deep Learning (DL) [12, 33], it is typical to learn an approximate prior $\mathcal{P}$ and then recover $x^*$ from $y$ by *approximate posterior sampling*, i.e., sampling $\hat{x}$ from the posterior $\mathcal{P}(\cdot|y, A)$. An increasingly popular approach involves using generative models to learn $\mathcal{P}$ (see, e.g., [12, 21, 33, 68, 70, 80] and references therein).

A major concern in many inverse problems is that the number of measurements $m$ is highly limited, due to physical constraints such as time (e.g., in Magnetic Resonance Imaging (MRI)), power (e.g., in portable sensors), money (e.g., seismic imaging), radiation exposure (e.g., X-Ray CT), or other factors [5, 64, 68]. Hence, one aims to recover $x^*$ well while keeping the number of measurements $m$ as small as possible. With this in mind, in this work we address the following broad question: *How many measurements suffice for stable and accurate recovery of $x^* \sim \mathcal{R}$ via approximate posterior*

39th Conference on Neural Information Processing Systems (NeurIPS 2025).

*sampling $\hat{x} \sim \mathcal{P}(\cdot|y, A)$, and what are conditions on $\mathcal{R}$, $\mathcal{P}$ and the distributions $\mathcal{A}$ and $\mathcal{E}$ of the measurement matrix $A$ and noise $e$, respectively, that ensure this recovery?*

## 1.1 Overview

In this work, we strive to answer to this question in the broadest possible terms, with a theoretical framework that allows for very general types of distributions. We now describe the corresponding conditions needed and present simplified versions of our main results.

**(i) Closeness of the real and approximate distributions.** We assume that $W_p(\mathcal{R}, \mathcal{P})$ is small for some $1 \leq p \leq \infty$, where $W_p$ denotes the Wassertstein $p$-metric.

**(ii) Low-complexity of $\mathcal{P}$.** Since $m \ll n$ in many applications, to have any prospect for accurate recovery we need to impose that $\mathcal{P}$ (or equivalently $\mathcal{R}$, in view of the previous assumption) has an inherent low complexity. Following [49], we quantify this in terms of its *approximate covering number* $\mathrm{Cov}_{\eta,\delta}(\mathcal{P})$. This is equal to the minimum number of balls of radius $\eta$ required to cover a region of $\mathbb{R}^n$ having $\mathcal{P}$-measure at least $1 - \delta$. See Definition 2.1 for the full definition.

**(iii) Concentration of $\mathcal{A}$.** We consider constants $C_{\mathsf{low}}(t) = C_{\mathsf{low}}(t; \mathcal{A}, D) \geq 0$ and $C_{\mathsf{upp}}(t) = C_{\mathsf{upp}}(t; \mathcal{A}, D) \geq 0$ (see Definition 2.3 for the full definition) such that
$$\mathbb{P}_{A\sim\mathcal{A}}[\|Ax\| \leq t\|x\|] \leq C_{\mathsf{low}}(t), \qquad \mathbb{P}_{A\sim\mathcal{A}}[\|Ax\| \geq t\|x\|] \leq C_{\mathsf{upp}}(t)$$
for all $x \in D := \mathrm{supp}(\mathcal{P}) - \mathrm{supp}(\mathcal{P})$ and $t > 0$. Here, and throughout this paper, we write $S - S = \{x_1 - x_2 : x_1, x_2 \in S\} \subseteq \mathbb{R}^n$ for the difference set associated with a set $S \subseteq \mathbb{R}^n$. We also write $\mathrm{supp}(\mathcal{P})$ for the support of the measure $\mathcal{P}$ (see §2.1 for the definition). Furthermore, we write $\|\cdot\|$ for the $\ell^2$-norm and $\|\cdot\|_\infty$ for the $\ell^\infty$-norm. If $\mathcal{A}$ is isotropic, i.e., $\mathbb{E}_{A\sim\mathcal{A}}\|Ax\|^2 = \|x\|^2$, $\forall x \in \mathbb{R}^n$, as is often the case in practice, these constants measure how fast $\|Ax\|^2$ concentrates around its mean. Notice that this condition is imposed only on $D = \mathrm{supp}(\mathcal{P}) - \mathrm{supp}(\mathcal{P})$, rather than the whole space $\mathbb{R}^n$. As we see later, this is crucial in obtaining meaningful recovery guarantees.

Finally, in order to present a simplified result in this section, we now make several simplifying assumptions. Both of these will be relaxed in our full result, Theorem 3.1.

**(iv) Gaussian noise.** Specifically, we assume that $\mathcal{E} = \mathcal{N}(0, \frac{\sigma^2}{m}I)$ for some $\sigma > 0$.

**(v) Bounded forwards operators.** We assume that $\|A\| \leq \theta$ a.s. for $A \sim \mathcal{A}$ and some $\theta > 0$.

**Theorem 1.1** (Simplified main result). *Let $1 \leq p \leq \infty$, $0 < \delta \leq 1/4$, $\varepsilon, \eta > 0$ and suppose that conditions (i)–(v) hold with $W_p(\mathcal{R}, \mathcal{P}) \leq \varepsilon/(2m\theta)$ and $\sigma \geq \varepsilon/\delta^{1/p}$. Suppose that $x^* \sim \mathcal{R}$, $A \sim \mathcal{A}$, $e \sim \mathcal{E}$ independently and $\hat{x} \sim \mathcal{P}(\cdot|y, A)$, where $y = Ax^* + e$. Then, for any $d \geq 2$,*

$$\mathbb{P}\left[\|x^* - \hat{x}\| \geq (8d^2 + 2)(\eta + \sigma)\right] \lesssim \delta + \mathrm{Cov}_{\eta,\delta}(\mathcal{P})\left[C_{\mathsf{low}}(1/d) + C_{\mathsf{upp}}(d) + \mathrm{e}^{-m/16}\right]. \quad (1.2)$$

Note that in this and all subsequent results, the term on the left-hand side of (1.2) is the probability with respect to all variables, i.e., $x^* \sim \mathcal{R}$, $A \sim \mathcal{A}$, $e \sim \mathcal{E}$ and $\hat{x} \sim \mathcal{P}(\cdot|y, A)$. Theorem 1.1 is extremely general, in that it allows for essentially arbitrary (real and approximate) signal distributions $\mathcal{R}$ and $\mathcal{P}$ and an essentially arbitrary distribution $\mathcal{A}$ for the forwards operator. In broad terms, it bounds the probability that the error $\|x^* - \hat{x}\|$ of posterior sampling exceeds a constant times the noise level $\sigma$ plus an arbitrary parameter $\eta$. It does so in terms of the approximate covering number $\mathrm{Cov}_{\eta,\delta}(\mathcal{P})$, which measures the complexity of the approximate distribution $\mathcal{P}$, the concentration bounds $C_{\mathsf{low}}$ and $C_{\mathsf{upp}}$ for $\mathcal{A}$, which measure how much $A$ elongates or shrinks a fixed vector, and an exponentially-decaying term $\mathrm{e}^{-m/16}$, which stems from the (Gaussian) noise. In particular, by analyzing these terms for different classes of distributions $\mathcal{P}$ and $\mathcal{A}$, we can derive concrete bounds for various exemplar problems. We next describe two such problems.

## 1.2 Examples

We first consider $\mathcal{A}$ to be a distribution of *subgaussian random matrices*. Here $A \sim \mathcal{A}$ if its entries are i.i.d. subgaussian random variables with mean zero and variance $1/m$ (see Definition 3.4).

**Theorem 1.2** (Subgaussian measurement matrices, simplified). *Consider the setup of Theorem 1.1, where $\mathcal{A}$ is a distribution of subgaussian random matrices. Then there is a constant $c > 0$ (depending on the subgaussian parameters $\beta, \kappa > 0$; see Definition 3.4) such that*

$$\mathbb{P}\left[\|x^* - \hat{x}\| \geq 34(\eta + \sigma)\right] \lesssim \delta, \quad \text{whenever } m \geq c \cdot \left[\log(\mathrm{Cov}_{\eta,\delta}(\mathcal{P})) + \log(1/\delta)\right].$$

This theorem shows the efficacy of Bayesian recovery with subgaussian random matrices: namely, the sample complexity scales *linearly* in the distribution complexity, i.e., the log of the approximate covering number. Later in Theorem 3.5, we slightly refine and generalize this result.

Gaussian random matrices are very commonly studied, due to their amenability to analysis and tight theoretical bounds [5, 22, 34, 49], with Theorem 1.2 being a case in point. However, they are largely irrelevant to practical inverse problems, where physical constraints impose certain structures on the forwards operator distribution $\mathcal{A}$ [5, 64, 68]. For instance, in MRI physical constraints mean that the measurements are samples of the Fourier transform of the image. This has motivated researchers to consider much more practically-relevant distributions, in particular, so-called *subsampled orthogonal transforms* (see, e.g., [5]). Here $U \in \mathbb{R}^{n \times n}$ is a fixed orthogonal matrix – for example, the matrix of the Discrete Fourier Transform (DFT) in the case of MRI – and the distribution $\mathcal{A}$ is defined by randomly selecting $m$ rows of $U$. See Definition 3.7 for the formal definition.

**Theorem 1.3** (Subsampled orthogonal transforms, simplified). *Consider the setup of Theorem 1.1, where $\mathcal{A}$ is a distribution of subsampled orthogonal matrices based on a matrix $U$. Then there is a universal constant $c > 0$ such that*

$$\mathbb{P}\left[\|x^* - \hat{x}\| \geq 34(\eta + \sigma)\right] \lesssim \delta, \quad \text{whenever } m \geq c \cdot \mu(U; D) \cdot \left[\log(\text{Cov}_{\eta,\delta}(\mathcal{P})) + \log(1/\delta)\right],$$

*where $D = \text{supp}(\mathcal{P}) - \text{supp}(\mathcal{P})$ and $\mu(U; D)$ is the coherence of $U$ relative to $D$, defined as*

$$\mu(U; D) = n \sup\left\{\|Ux\|_\infty^2 / \|x\|^2 : x \in D, \ x \neq 0\right\}.$$

This result shows that similar stable and accurate recovery to the Gaussian case can be achieved using subsampled orthogonal matrices, provided the number of measurements scales linearly with the coherence. We discuss this term further in §3.2, where we also present the full result, Theorem 3.9.

In general, Theorems 1.2 and 1.3 establish a key condition for successful Bayesian recovery in inverse problems, in each case relating the number of measurements $m$ to the intrinsic complexity $\log(\text{Cov}_{\eta,\delta}(\mathcal{P}))$ of $\mathcal{P}$. It is therefore informative to see how this complexity behaves for cases of interest. As noted, it is common to use a generative model to learn $\mathcal{P}$. This means that $\mathcal{P} = G\sharp\gamma$, where $G : \mathbb{R}^k \to \mathbb{R}^n$ is a Deep Neural Network (DNN) and $\gamma$ is some fixed probability measure on the latent space $\mathbb{R}^k$. Typically, $\gamma = \mathcal{N}(0, I)$. If $G$ is $L$-Lipschitz, we show in Proposition 4.1 that

$$\log(\text{Cov}_{\eta,\delta}(\mathcal{P})) = \mathcal{O}(k \log[L(\sqrt{k} + \log(1/\delta))/\eta]). \tag{1.3}$$

This scales log-linearly in the latent space dimension $k$, confirming the intrinsic low-complexity of $\mathcal{P}$. Combining (1.3) with Theorems 1.2-1.3 we see that posterior sampling achieves stable and accurate recovery, provided the number of measurements $m$ scales near-optimally with $k$, i.e., as $\mathcal{O}(k \log(k))$.

Further, in order to compare with deterministic settings such as classical compressed sensing, which concerns the recovery of $s$-sparse vectors, we also consider distributions $\mathcal{P} = \mathcal{P}_s$ of $s$-sparse vectors. In this case, we show in Proposition 4.3 that

$$\log(\text{Cov}_{\eta,\delta}(\mathcal{P})) = \mathcal{O}(s \log(n/s) + \log[(\sqrt{s} + \log(1/\delta))/\eta]). \tag{1.4}$$

Hence $s$ measurements, up to log terms, are sufficient for recovery of approximately sparse vectors. This extends a classical result for deterministic compressed sensing to the Bayesian setting.

## 1.3 Significance

The significance of this work is as follows. See §1.4 for additional discussion.

**1.** We provide the first results for Bayesian recovery with arbitrary real and approximate prior distributions $\mathcal{R}, \mathcal{P}$ and forwards operator and noise distributions $\mathcal{A}, \mathcal{E}$.

**2.** Unlike much of the theory of Bayesian inverse problems, which is *asymptotic* in nature [12, 30, 71], our results are *non-asymptotic*. They hold for arbitrary values of the various parameters within given ranges (e.g., the failure probability $\delta$, the noise level $\sigma$, the number of measurements $m$, and so forth).

**3.** For priors defined by Lipschitz generative DNNs, we establish the first result demonstrating that the sample complexity of Bayesian recovery depends log-linearly on the latent dimension $k$ and logarithmically on the Lipschitz constant.

**4.** For the important class of subsampled orthogonal transforms, we show that the sample complexity of Bayesian recovery depends on the *coherence*, thus resolving several key open problems in the literature (see next).

**5.** It is increasingly well-known that DL-based methods for inverse problems are susceptible to *hallucinations* and other undesirable effects [11, 15, 18, 25, 29, 40, 42, 45, 47, 60–63, 65, 66, 79]. This is a major issue that may limit the uptake of these methods in safety-critical domains such as medical imaging [25, 55, 57, 61, 74, 76, 77]. Our results provide theoretical guarantees for stable and accurate, and therefore show conditions under which hallucinations provably cannot occur. This is not only theoretically interesting, but it also has practical consequences in the development of robust DL methods for inverse problems – a topic we intend to explore in future work.

## 1.4 Related work

Bayesian methods for inverse problems have become increasingly popular over the last several decades [30, 71], and many state-of-the-art DL methods for inverse problems now follow a Bayesian approach (see [7, 12, 21, 46, 56, 64] and references therein). Learned priors, such as those stemming from generative models, are now also increasingly used in applications [1, 7, 12, 21, 28, 33, 38, 46, 48, 51, 53, 54, 58, 64, 68, 70, 78, 80].

This work is motivated in part by the (non-Bayesian) theory of generative models for solving inverse problems (see [22, 33, 43, 44, 48, 68–70, 80] and references therein). This was first developed in [22], where compressed sensing techniques were used to show recovery guarantees for a Gaussian random matrix $A$ when computing an approximate solution to (1.1) in the range $\Sigma := \mathrm{ran}(G)$ of a Lipschitz map $G : \mathbb{R}^k \to \mathbb{R}^n$, typically assumed to be a generative DNN. This is a *deterministic* approach. Besides the random forward operator and (potentially) random noise, it recovers a fixed (i.e., nonrandom) underlying signal $x^*$ in a deterministic fashion with a *point estimator* $\hat{x}$ that is obtained as a minimizer of the empirical $\ell^2$-loss $\min_{z \in \Sigma} \|Az - y\|^2$. In particular, no information about the latent space distribution $\gamma$ is used. In this work, following, e.g., [1, 21, 48, 49, 70] and others, we consider a Bayesian setting, where $x^* \sim \mathcal{R}$ is random and where we quantify the number of measurements that suffice for accurate and stable recovery via posterior sampling $\hat{x} \sim \mathcal{P}(\cdot|y, A)$.

Our work is a generalization of [49], which considered Bayesian recovery with Gaussian random matrices and standard Gaussian noise. We significantly extend [49] to allow for arbitrary distributions $\mathcal{A}$ and $\mathcal{E}$ for the forward operator and noise, respectively. A key technical step in doing this is the introduction of the concentration bounds $C_{\mathsf{low}}$ and $C_{\mathsf{upp}}$. In particular, these bounds are imposed only over the subset $D = \mathrm{supp}(\mathcal{P}) - \mathrm{supp}(\mathcal{P})$. This is unnecessary in the Gaussian case considered in Theorem 1.2, but crucial to obtain meaningful bounds in, for instance, the case of subsampled orthogonal transforms considered in Theorem 1.3 (see Remarks 3.2 and 3.10 for further discussion). As noted, this case is very relevant to applications. In particular, when $U$ is taken as a DFT matrix our work addresses open problems posed in [48, §3] and [68, §II.F] on recovery guarantees with Fourier measurements. We also derive bounds for the approximate covering number of distributions given by Lipschitz generative DNNs (see (1.3) and Proposition 4.1) and distributions of sparse vectors (see (1.4) and Proposition 4.3), addressing an open problem posed in [49, §6]. In particular, we demonstrate stable and accurate recovery in a Bayesian sense with a number of measurements that scales linearly in the model complexity, i.e., $k$ in the former case and $s$ in the latter case.

Recently [16, 17] generalized the results of [22] in the non-Bayesian setting from Gaussian random matrices to subsampled orthogonal transforms. Theorem 1.3 provides a Bayesian analogue of this work, as discussed above, where we consider posterior sampling rather than a deterministic point estimator. We also extend the setup of [16, 17] by allowing for general measurement distributions $\mathcal{A}$ and priors $\mathcal{P}$. In particular, in [16, 17] the quantity $\Sigma$, which is the deterministic analogue of the prior distribution $\mathcal{P}$ in our work, was assumed to be the range of a ReLU generative DNN. Like in [16], we make use of the concept of coherence (see Theorem 1.3). However, our proof techniques are completely different to those of [16, 17].

Classical compressed sensing considers the recovery of approximately $s$-sparse vectors from (1.1). However, it has been extended to consider much more general types of *low-complexity* signal models, such as joint or block sparse vectors, tree sparse vectors, cosparse vectors and many others [4, 6, 13, 23, 31, 36, 73]. However, most recovery guarantees for general model classes consider only (sub)Gaussian measurement matrices (see e.g., [13, 34]). Recently, [3] introduced a general

framework for compressed sensing that allows for general low-complexity models $\Sigma \subseteq \mathbb{R}^n$ contained within a finite union of finite-dimensional subspaces *and* arbitrary (random) measurement matrices. Our work is a Bayesian analogue of this deterministic setting. Similar to [3], a key feature of our work is that we consider arbitrary real and approximate signal distributions $\mathcal{P}, \mathcal{R}$ (analogous to arbitrary low-complexity models $\Sigma$) and arbitrary distributions $\mathcal{A}$ for the forwards operator. Unsurprisingly, a number of the conditions in our main result, Theorem 1.1 – namely, the low-complexity condition (ii) and concentration condition (iii) – share similarities to those that ensure stable and accurate recovery in non-Bayesian compressed sensing. See Remarks 2.2 and 3.3 for further discussion. However, the proof techniques used in this work are once more entirely different.

Finally, while the focus of this work is to establish guarantees for posterior sampling, we mention in passing related work on information-theoretically optimal recovery methods. Methods such as Approximate Message Passing (AMP) [14, 35] are well studied, and asymptotic information-theoretic bounds for Gaussian random matrices are known [52]. AMP methods are fast point estimation algorithms, whereas we focus on sampling-based methods and do not consider computational implementations (see §5 for some further discussion). Note that information-theoretic lower bounds for posterior sampling have also been shown for the Gaussian case in [49].

## 2 Preliminaries

### 2.1 Notation

We now introduce some further notation. We let $B_r(x) = \{z \in \mathbb{R}^n : \|z - x\| \leq r\}$ and, when $x = 0$, we write $B_r := B_r(0)$. Given a set $X \subseteq \mathbb{R}^n$, we write $X^c = \mathbb{R}^n \backslash X$ for its complement. We also write $B_r(X) = \bigcup_{x \in X} B_r(x)$ for the $r$-neighbourhood of $X$.

Let $(X, \mathcal{F}, \mu)$ be a Borel probability space. We write $\mathrm{supp}(\mu)$ for its support, i.e., the smallest closed set $A \subseteq X$ for which $\mu(A) = 1$. Given probability spaces $(X, \mathcal{F}_1, \mu), (Y, \mathcal{F}_2, \nu)$, we write $\Gamma = \Gamma_{\mu,\nu}$ for the set of couplings, i.e., probability measures on the product space $(X \times Y, \sigma(\mathcal{F}_1 \otimes \mathcal{F}_2))$ whose marginals are $\mu$ and $\nu$, respectively. Given a cost function $c : X \times Y \to [0, \infty)$ and $1 \leq p < \infty$, the Wasserstein-$p$ metric is defined as

$$W_p(\mu, \nu) = \inf_{\gamma \in \Gamma} \left( \int_{X \times Y} c(x, y)^p d\gamma(x, y) \right)^{1/p}.$$

If $p = \infty$, then $W_\infty(\mu, \nu) = \inf_{\gamma \in \Gamma}(\mathrm{esssup}_\gamma c(x, y))$. In this paper, unless stated otherwise, $X = Y = \mathbb{R}^n$ and the cost function $c$ is the Euclidean distance.

### 2.2 Approximate covering numbers

As a measure of complexity of measures, we use the concept of approximate covering numbers as introduced in [49].

**Definition 2.1** (Approximate covering number). Let $(X, \mathcal{F}, \mathcal{P})$ be a probability space and $\delta, \eta \geq 0$. The $\eta, \delta$-*approximate covering number of* $\mathcal{P}$ is defined as

$$\mathrm{Cov}_{\eta,\delta}(\mathcal{P}) = \min \left\{ k \in \mathbb{N} : \exists \{x_i\}_{i=1}^k \subseteq \mathrm{supp}(\mathcal{P}), \ \mathcal{P} \left( \bigcup_{i=1}^k B_\eta(x_i) \right) \geq 1 - \delta \right\}.$$

This quantity measures how many balls of radius $\eta$ are required to cover at least $1 - \delta$ of the $\mathcal{P}$-mass of $\mathbb{R}^n$. See [49] for further discussion. Note that [49] does not require the centres $x_i$ of the approximate cover belong to $\mathrm{supp}(\mathcal{P})$. However, this is useful in our more general setting and presents no substantial restriction. At worst, this requirement changes $\eta$ by a factor of $1/2$.

**Remark 2.2 (Relation to non-Bayesian compressed sensing)** Note that when $\delta = 0$, the approximate covering number $\mathrm{Cov}_{\eta,0}(\mathcal{P}) \equiv \mathrm{Cov}_\eta(\mathrm{supp}(\mathcal{P}))$ is just the classical covering number of the set $\mathrm{supp}(\mathcal{P})$, i.e., the minimal number of balls of radius $\eta$ that cover $\mathrm{supp}(\mathcal{P})$. Classical covering numbers play a key role in (non-Bayesian) compressed sensing theory. Namely, the covering number of (the unit ball of) the model class $\Sigma \subseteq \mathbb{R}^n$ directly determines the number of measurements that suffice for stable and accurate recovery. See, e.g., [3, 34]. In the Bayesian setting, the approximate covering number plays the same role; see Theorem 1.1.

## 2.3 Bounds for $\mathcal{A}$ and $\mathcal{E}$

Since our objective is to establish results that hold for arbitrary measurement and noise distributions $\mathcal{A}$ and $\mathcal{E}$, we require several key definitions. These are variety of (concentration) bounds.

**Definition 2.3** (Concentration bounds for $\mathcal{A}$). Let $\mathcal{A}$ be a distribution on $\mathbb{R}^{m \times n}$, $t \geq 0$ and $D \subseteq \mathbb{R}^n$. Then a *lower concentration bound for $\mathcal{A}$* is any constant $C_{\mathsf{low}}(t) = C_{\mathsf{low}}(t; \mathcal{A}, D) \geq 0$ such that

$$\mathbb{P}_{A \sim \mathcal{A}}\{\|Ax\| \leq t\|x\|\} \leq C_{\mathsf{low}}(t; \mathcal{A}, D), \quad \forall x \in D.$$

Similarly, an *upper concentration bound for $\mathcal{A}$* is any constant $C_{\mathsf{upp}}(t) = C_{\mathsf{upp}}(t; \mathcal{A}, D) \geq 0$ such that

$$\mathbb{P}_{A \sim \mathcal{A}}\{\|Ax\| \geq t\|x\|\} \leq C_{\mathsf{upp}}(t; \mathcal{A}, D), \quad \forall x \in D.$$

Finally, given $t, s \geq 0$ an *(upper) absolute concentration bound for $\mathcal{A}$* is any constant $C_{\mathsf{abs}}(s, t; \mathcal{A}, D)$ such that

$$\mathbb{P}_{A \sim \mathcal{A}}(\|Ax\| > t) \leq C_{\mathsf{abs}}(s, t; \mathcal{A}, D), \quad \forall x \in D, \ \|x\| \leq s.$$

Notice that if $\mathcal{A}$ is isotropic, i.e., $\mathbb{E}\|Ax\|^2 = \|x\|^2$, $\forall x \in \mathbb{R}^n$, then $C_{\mathsf{low}}$ and $C_{\mathsf{upp}}$ determine how well $\|Ax\|^2$ concentrates around its mean $\|x\|^2$ for any fixed $x \in D$. To obtain desirable sample complexity estimates (e.g., Theorems 1.2 and 1.3), we need concentration bounds that decay exponentially in $m$. A crucial component of this analysis is considering concentration bounds over some subset $D$ (related to the support of $\mathcal{P}$), as, in general, one cannot expect fast concentration over the whole of $\mathbb{R}^n$. See Remarks 3.2 and 3.10.

**Definition 2.4** (Concentration bound for $\mathcal{E}$). Let $\mathcal{E}$ be a distribution in $\mathbb{R}^m$ and $t \geq 0$. Then an *(upper) concentration bound for $\mathcal{E}$* is any constant $D_{\mathsf{upp}}(t) = D_{\mathsf{upp}}(t; \mathcal{E}) \geq 0$ such that

$$\mathcal{E}(B_t^c) = \mathbb{P}_{e \sim \mathcal{E}}(\|e\| \geq t) \leq D_{\mathsf{upp}}(t; \mathcal{E}).$$

Notice that this bound just measures the probability that the noise is large. We also need the following concept, which estimates how much the density of $\mathcal{E}$ changes in a $\tau$-neighbourhood of the origin when perturbed by an amount $\varepsilon$.

**Definition 2.5** (Density shift bounds for $\mathcal{E}$). Let $\mathcal{E}$ be a distribution in $\mathbb{R}^m$ with density $p_{\mathcal{E}}$ and $\varepsilon, \tau \geq 0$. Then a *density shift bound for $\mathcal{E}$* is any constant $D_{\mathsf{shift}}(\varepsilon, \tau) = D_{\mathsf{shift}}(\varepsilon, \tau; \mathcal{E}) \geq 0$ (possibly $+\infty$) such that

$$p_{\mathcal{E}}(u) \leq D_{\mathsf{shift}}(\varepsilon, \tau; \mathcal{E}) p_{\mathcal{E}}(v), \quad \forall u, v \in \mathbb{R}^n, \ \|u\| \leq \tau, \ \|u - v\| \leq \varepsilon.$$

# 3 Main results

We now present our main results. The first, an extension of Theorem 1.1, is a general result that holds for arbitrary distributions $\mathcal{R}, \mathcal{P}, \mathcal{A}$ and $\mathcal{E}$.

**Theorem 3.1.** *Let $1 \leq p \leq \infty$, $0 \leq \delta \leq 1/4$, $\varepsilon, \eta, t > 0$, $c, c' \geq 1$ and $\sigma \geq \varepsilon/\delta^{1/p}$. Let $\mathcal{E}$ be a distribution on $\mathbb{R}^m$ and $\mathcal{R}, \mathcal{P}$ be distributions on $\mathbb{R}^n$ satisfying $W_p(\mathcal{R}, \mathcal{P}) \leq \varepsilon$ and*

$$\min(\log \mathrm{Cov}_{\eta, \delta}(\mathcal{R}), \log \mathrm{Cov}_{\eta, \delta}(\mathcal{P})) \leq k \tag{3.1}$$

*for some $k \in \mathbb{N}$. Suppose that $x^* \sim \mathcal{R}$, $A \sim \mathcal{A}$, $e \sim \mathcal{E}$ independently and $\hat{x} \sim \mathcal{P}(\cdot|y, A)$, where $y = Ax^* + e$. Then $p := \mathbb{P}[\|x^* - \hat{x}\| \geq (c+2)(\eta + \sigma)]$ satisfies*

$$p \leq 2\delta + C_{\mathsf{abs}}(\varepsilon/\delta^{1/p}, t\varepsilon/\delta^{1/p}; \mathcal{A}, D_1) + D_{\mathsf{upp}}(c'\sigma; \mathcal{E})$$

$$+ 2D_{\mathsf{shift}}(t\varepsilon/\delta^{1/p}, c'\sigma; \mathcal{E})\mathrm{e}^k \left[ C_{\mathsf{low}}\left(\frac{2\sqrt{2}}{\sqrt{c}}; \mathcal{A}, D_2\right) + C_{\mathsf{upp}}\left(\frac{\sqrt{c}}{2\sqrt{2}}; \mathcal{A}, D_2\right) + 2D_{\mathsf{upp}}\left(\frac{\sqrt{c}\sigma}{2\sqrt{2}}; \mathcal{E}\right) \right],$$

*where*

$$D_1 = B_{\varepsilon/\delta^{1/p}}(\mathrm{supp}(\mathcal{P})) \cap \mathrm{supp}(\mathcal{R}) - \mathrm{supp}(\mathcal{P}) \tag{3.2}$$

*and*

$$D_2 = \begin{cases} \mathrm{supp}(\mathcal{P}) - \mathrm{supp}(\mathcal{P}) & \text{if } \mathcal{P} \text{ attains the minimum in (3.1)} \\ \mathrm{supp}(\mathcal{P}) - \mathrm{supp}(\mathcal{R}) & \text{otherwise} \end{cases}. \tag{3.3}$$

This theorem bounds the probability $p$ of unstable or inaccurate recovery in terms of the various parameters using the constants introduced in the previous section and the approximate covering numbers of $\mathcal{R}, \mathcal{P}$. This result is powerful in its generality, but as a consequence, rather opaque. In particular, since it considers arbitrary measurement and noise distributions, the number of measurements $m$ does not explicitly enter the bound. For typical distributions, a dependence on $m$ is found in the concentration bounds $C_{\mathsf{low}}, C_{\mathsf{upp}}, D_{\mathsf{upp}}$, as well as the terms $D_{\mathsf{shift}}$ and $C_{\mathsf{abs}}$. For instance, the former decay exponentially-fast in $m$ for the examples introduced in §1.2, and therefore compensate for the exponentially-large scaling in $k$ in the main bound (see §B for precise estimates, as well as the discussion in §3.1-3.2). Note that Theorem 3.1 also considers general noise distributions $\mathcal{E}$. While Gaussian noise is arguably the most important example – and will be used in all our subsequent examples – this additional generality comes at little cost in terms of the technicality of the proofs.

**Remark 3.2 (The concentration bounds in Theorem 3.1)** A particularly important facet of this result, for the reasons discussed above, is that the various concentration bounds $C_{\mathsf{abs}}, C_{\mathsf{low}}$ and $C_{\mathsf{upp}}$ are taken over sets $D_1, D_2$ – given by (3.2) and (3.3), respectively, and related to the support of $\mathcal{P}$ and $\mathcal{R}$ – rather than the whole space $\mathbb{R}^n$. We exploit this fact crucially later in Theorem 3.9.

**Remark 3.3 (Relation to non-Bayesian compressed sensing)** The constants $C_{\mathsf{low}}$ and $C_{\mathsf{upp}}$ are similar, albeit not identical to similar conditions such as the Restricted Isometry Property (RIP) (see, e.g., [5, Chpt. 5]) or Restricted Eigenvalue Condition (REC) [19, 22] that appear in non-Bayesian compressed sensing. There, one considers a fixed model class $\Sigma \subseteq \mathbb{R}^n$, such as the set $\Sigma_s$ of $s$-sparse vectors or, as in [22], the range $\mathrm{ran}(G)$ of a generative DNN. Conditions such as the RIP or REC impose that $\|Ax\|$ is concentrated around $\|x\|$ for all $x$ belonging to the difference set $\Sigma - \Sigma$. In Theorem 3.1, assuming $\mathcal{P}$ attains the minimum in (3.1), there is a similar condition with $\Sigma$ replaced by $\mathrm{supp}(\mathcal{P})$. Indeed, $C_{\mathsf{low}}(2\sqrt{2}/\sqrt{c}; \mathcal{A}, D_2)$ measures how small $\|Ax\|$ is in relation to $\|x\|$ and $C_{\mathsf{upp}}(\sqrt{c}/(2\sqrt{2}); \mathcal{A}, D_2)$ measures how large $\|Ax\|$ is in relation to $\|x\|$.

## 3.1 Example: Subgaussian random matrices with Gaussian noise

We now apply this theorem to the first example introduced in §1.2. Recall that a random variable $X$ on $\mathbb{R}$ is *subgaussian with parameters* $\beta, \kappa > 0$ if $\mathbb{P}(|X| \geq t) \leq \beta \mathrm{e}^{-\kappa t^2}$ for all $t > 0$.

**Definition 3.4** (Subgaussian random matrix). A random matrix $A \in \mathbb{R}^{m \times n}$ is *subgaussian with parameters* $\beta, \kappa > 0$ if $A = \frac{1}{\sqrt{m}} \widetilde{A}$, where the entries of $\widetilde{A}$ are independent mean-zero sugaussian random variables with variance 1 and the same subgaussian parameters $\beta, \kappa$.

Note that $1/\sqrt{m}$ is a scaling factor that ensures that $A$ is *isotropic*, i.e., $\mathbb{E}\|Ax\|^2 = \|x\|^2, \forall x \in \mathbb{R}^n$.

**Theorem 3.5.** *Let $1 \leq p \leq \infty$, $0 \leq \delta \leq 1/4$, $\varepsilon, \eta > 0$ and $\sigma \geq \varepsilon/\delta^{1/p}$. Let $\mathcal{E} = \mathcal{N}(0, \frac{\sigma^2}{m} I)$ and $\mathcal{A}$ be a distribution of subgaussian random matrices with parameters $\beta, \kappa > 0$. Let $\mathcal{R}, \mathcal{P}$ be distributions on $\mathbb{R}^n$ and suppose that $x^* \sim \mathcal{R}$, $A \sim \mathcal{A}$, $e \sim \mathcal{E}$ independently and $\hat{x} \sim \mathcal{P}(\cdot|y, A)$, where $y = Ax^* + e$. Then there is a constant $c(\beta, \kappa) > 0$ depending on $\beta, \kappa$ only such that*

$$\mathbb{P}[\|x^* - \hat{x}\| \geq 34(\eta + \sigma)] \lesssim \delta,$$

*provided $W_p(\mathcal{R}, \mathcal{P}) \leq \varepsilon/c(\beta, \kappa)$ and*

$$m \geq c(\beta, \kappa) \cdot [\min(\log \mathrm{Cov}_{\eta, \delta}(\mathcal{R}), \log \mathrm{Cov}_{\eta, \delta}(\mathcal{P})) + \log(1/\delta)]. \tag{3.4}$$

This theorem is derived from Theorem 3.1 by showing that the various concentration bounds are exponentially small in $m$ for subgaussian random matrices (see §B). It is a direct generalization of [49], which considered the Gaussian case only. It shows that subgaussian random matrices are near-optimal for Bayesian recovery, in the sense that $m$ scales linearly with the log of the approximate covering number (3.4). We estimate these covering numbers for several key cases in §4.

It is worth at this stage discussing how (3.4) behaves with respect to the various parameters. First, suppose that $\eta$ decreases so that the error bound becomes smaller. Then $\mathrm{Cov}_{\eta, \delta}(\cdot)$ increases, meaning, as expected, that more measurements are required to meet (3.4). Second, suppose that $\delta$ decreases, so that the failure probability shrinks. Then $\mathrm{Cov}_{\eta, \delta}(\cdot)$ and $\log(1/\delta)$ both increase, meaning, once again, that more measurements are needed for (3.4) to hold. Both behaviours are as expected.

**Remark 3.6 (Relation to Johnson–Lindenstrauss)** Suppose that $\mathcal{P} = \mathcal{R}$ is a sum of $d$ Diracs located at $X = \{x_1, \ldots, x_d\} \subseteq \mathbb{R}^n$. Since the matrix $A \in \mathbb{R}^{m \times n}$ is a linear dimensionality-reducing map, the Johnson-Lindenstrauss Lemma states that distances in $X$ are preserved under $A$ if and only if $m = O(\log(d))$. In this setting, preserving the distances in $X$ is equivalent to being able to stably identify the mode from which a signal is drawn when observing its measurements. In agreement with this argument, Theorem 3.5 also predicts recovery from roughly $m = O(\log(d))$ measurements.

## 3.2 Example: Randomly-subsampled orthogonal transforms with Gaussian noise

As discussed, subgaussian random matrices are largely impractical. We now consider the more practical case of subsampled orthogonal transforms.

**Definition 3.7** (Randomly-subsampled orthogonal transform)**.** Let $U \in \mathbb{R}^{n \times n}$ be orthogonal (i.e., $U^\top U = UU^\top = I$) and write $u_1, \ldots, u_n \in \mathbb{R}^n$ for its rows. Let $X_1, \ldots, X_n \sim_{\text{i.i.d.}} \text{Ber}(m/n)$ be independent Bernoulli random variables with $\mathbb{P}(X_i = 1) = m/n$ and $\mathbb{P}(X_i = 0) = 1 - m/n$. Then we define a distribution $\mathcal{A}$ as follows. We say that $A \sim \mathcal{A}$ if

$$A = \sqrt{\frac{n}{m}} \begin{bmatrix} u_{i_1}^\top \\ \vdots \\ u_{i_q}^\top \end{bmatrix},$$

where $\{i_1, \ldots, i_q\} \subseteq \{1, \ldots, n\}$ is the set of indices $i$ for which $X_i = 1$.

The factor $\sqrt{n/m}$ ensures that $\mathbb{E}(A^\top A) = I$. Note that the number of measurements $q$ in this model is itself a random variable, with $\mathbb{E}(q) = m$. However, $q$ concentrates exponentially around its mean.

**Definition 3.8.** Let $U \in \mathbb{R}^{n \times n}$ and $D \subseteq \mathbb{R}^n$. The *coherence of $U$ relative to $D$* is

$$\mu(U; D) = n \cdot \sup \left\{ \|Ux\|_\infty^2 / \|x\|^2 : x \in D, \ x \neq 0 \right\}.$$

Coherence is a well-known concept in classical compressed sensing with sparse vectors. Definition 3.8 is a generalization that allows for arbitrary model classes $D$. This definition is similar to that of [16], which considered non-Bayesian compressed sensing with generative models. It is also related to the more general concept of *variation* introduced in [3].

**Theorem 3.9.** *Let $1 \leq p \leq \infty$, $0 \leq \delta \leq 1/4$, $\varepsilon, \eta > 0$ and $\sigma \geq \varepsilon/\delta^{1/p}$. Let $\mathcal{E} = \mathcal{N}(0, \frac{\sigma^2}{m}I)$ and $\mathcal{A}$ be a distribution of randomly-subsampled orthogonal matrices based on a matrix $U$. Let $\mathcal{R}, \mathcal{P}$ be distributions on $\mathbb{R}^n$ and suppose that $x^* \sim \mathcal{R}$, $A \sim \mathcal{A}$, $e \sim \mathcal{E}$ independently and $\hat{x} \sim \mathcal{P}(\cdot|y, A)$, where $y = Ax^* + e$. Then there is a universal constant $c > 0$ such that*

$$\mathbb{P}\left[\|x^* - \hat{x}\| \geq 34(\eta + \sigma)\right] \lesssim \delta,$$

*provided $W_p(\mathcal{R}, \mathcal{P}) \leq \varepsilon/(2\sqrt{mn})$ and*

$$m \geq c \cdot \mu(U; D) \cdot \left[\log(\text{Cov}_{\eta,\delta}(\mathcal{P})) + \log(1/\delta)\right], \quad \textit{where } D = \text{supp}(\mathcal{P}) - \text{supp}(\mathcal{P}). \quad (3.5)$$

This theorem is a Bayesian analogue of the deterministic results shown in [3, 16]. In [3, 16], the measurement conditions scale linearly with $\mu(U; D)$, where $D = \Sigma - \Sigma$ and $\Sigma \subseteq \mathbb{R}^n$ is the low-complexity model class. Similarly, the number of measurements (3.5) scales linearly with respect to the coherence relative to $D = \text{supp}(\mathcal{P}) - \text{supp}(\mathcal{P})$, which, as discussed in Remark 3.3, plays the role of the low-complexity model class in the Bayesian setting. Note that the measurement condition (3.5) involves the approximate distribution $\mathcal{P}$ only. This is relevant, since the quantities $\mu(U; D)$ and $\text{Cov}_{\eta,\delta}(\mathcal{P})$ can be estimated either numerically or analytically in various cases, such as when $\mathcal{P}$ is given by a generative model. Indeed, we estimate $\text{Cov}_{\eta,\delta}(\mathcal{P})$ analytically for Lipschitz generative models in Proposition 4.1 below. The coherence $\mu(U; D)$ was estimated analytically in [16] for ReLU DNNs with random weights (see Remark 4.2 below). It can also be estimated numerically for more general types of generative models [2, 16]. Overall, by estimating these quantities, one can use (3.5) to gauge how well one can recover with a given $\mathcal{P}$. Note that this may not be possible if (3.5) involved $\mathcal{R}$ as well, since this distribution is typically unknown.

In classical compressed sensing, coherence determines the sample complexity of recovering sparse vectors from randomly-subsampled orthogonal transforms [26]. A similar argument can be made in the Bayesian setting. Notice that $\mu(U; D) \leq \mu(U; \mathbb{R}^n) = n$. However, we are particularly interested in cases where $\mu(U; D) \ll n$, in which case (3.5) may be significantly smaller than the ambient dimension $n$. We discuss this in the context of several examples in the next section.

**Remark 3.10 (Concentration over subsets)** Theorem 3.9 illustrates why it is important that Theorem 3.1 involves concentration bounds over subsets of $\mathbb{R}^n$. To derive Theorem 3.9 from Theorem 3.1 (see §B), we show exponentially-fast concentration in $m/\mu(U; D)$. Had we considered the whole of $\mathbb{R}^n$, then, since $\mu(U; \mathbb{R}^n) = n$, this would have lead to an undesirable measurement condition of the form $m = \mathcal{O}(n)$ scaling linearly in the ambient dimension $n$.

## 4 Covering number and sample complexity estimates

We conclude by applying our results to two different types of approximate prior distributions.

### 4.1 Generative DNNs

**Proposition 4.1** (Approximate covering number for a Lipschitz pushforward of a Gaussian measure). *Let $G : \mathbb{R}^k \to \mathbb{R}^n$ be Lipschitz with constant $L \geq 0$, i.e., $\|G(x) - G(z)\| \leq L\|x - z\|$, $\forall x, z \in \mathbb{R}^k$, and define $\mathcal{P} = G\sharp\gamma$, where $\gamma = \mathcal{N}(0, I)$ is the standard normal distribution on $\mathbb{R}^k$. Then*

$$\log(\mathrm{Cov}_{\eta,\delta}(\mathcal{P})) \leq k \log\left[1 + \frac{2\sqrt{k}L}{\eta}\left(1 + \sqrt{\frac{2}{k}\log(1/\delta)}\right)\right]. \tag{4.1}$$

This result shows that $\mathcal{P}$ has low complexity, since $\log(\mathrm{Cov}_{\eta,\delta}(\mathcal{P}))$ scales log-linearly in $k$. Combined with Theorem 3.5, it shows that accurate and stable Bayesian recovery with such a prior with a sample complexity that is near-optimal in the latent dimension $k$, i.e., $\mathcal{O}(k \log(k))$.

Notice that $L$ only appears logarithmically in (4.1). While it is not the main focus of this work, we note that Lipschitz constants of DNNs have been studied quite extensively [37, 72, 75]. Moreover, it is also possible to design and train DNNs with small Lipschitz constants [59].

**Remark 4.2 (Quadratic bottleneck and high coherence)** In Theorem 3.9, the measurement condition (3.5) also depends on the coherence. This quantity has been considered in [16] for the case of ReLU DNNs. In particular, if a ReLU DNN $G : \mathbb{R}^k \to \mathbb{R}^n$ has random weights drawn from a standard normal distribution, then its coherence $\mu(U; D)$ scales like $\mathcal{O}(k)$ up to log factors [16, Thm. 3]. Combining this with Theorem 3.9 and Proposition 4.1, we see that the overall sample complexity for Bayesian recovery scales like $\mathcal{O}(k^2 \log(k))$ in this case. This is worse than the subgaussian case, where there is no coherence factor and the sample complexity, as noted above, is $\mathcal{O}(k \log(k))$. Such a *quadratic bottleneck* also arises in the non-Bayesian setting [16]. Its removal is an open problem (see §5). Note that the coherence is also not guaranteed to be small for general (in particular, trained) DNNs. However, [16] also discuss strategies for training generative models to have small coherence. Numerically, they show that generative models with smaller coherence achieve better recovery from the same number of measurements than those with larger coherence.

### 4.2 Distributions of sparse vectors

Let $s \in \{1, \ldots, n\}$. We define a distribution $\mathcal{P} = \mathcal{P}_s$ of $s$-sparse vectors in $\mathbb{R}^n$ as follows. To draw $x \sim \mathcal{P}$, we first choose a support set $S \subseteq \{1, \ldots, n\}$, $|S| = s$, uniformly at random amongst all possible $\binom{n}{s}$ such subsets. We then define $x_i = 0$, $i \notin S$, and for each $i \in S$ we draw $x_i$ randomly and independently from $\mathcal{N}(0, 1)$. Note that there are other ways to define distributions of sparse vectors, which can be analyzed similarly. However, for brevity we only consider the above setup.

**Proposition 4.3** (Approximate covering number for distributions of sparse vectors). *Let $\mathcal{P} = \mathcal{P}_s$ be a distribution of $s$-sparse vectors in $\mathbb{R}^n$. Then*

$$\log(\mathrm{Cov}_{\eta,\delta}(\mathcal{P}_s)) \leq s\left[\log\left(\frac{en}{s}\right) + \log\left(1 + \frac{2\sqrt{s}}{\eta}\left(1 + \sqrt{\frac{2}{s}\log(1/\delta)}\right)\right)\right]. \tag{4.2}$$

As in the previous case, we deduce Bayesian recovery from $\mathcal{O}(s \log(n/s))$ subgaussian measurements, i.e., near-optimal, log-linear sample complexity. In the case of randomly-subsampled orthogonal matrices, we also need to consider the coherence. For $\mathcal{P} = \mathcal{P}_s$ as above, one can easily show that

$$\mu(U; \mathrm{supp}(\mathcal{P}_s) - \mathrm{supp}(\mathcal{P}_s)) \leq 2s\mu_*(U), \qquad \text{where } \mu_*(U) = n \cdot \max_{i,j}|u_{ij}|^2. \tag{4.3}$$

The term $\mu_*(U)$ is often referred to as the *coherence of $U$* (see, e.g., [5, Defn. 5.8] or [26]). Notice that $\mu_*(U) \approx 1$ for DFT matrices, which is one reason why subsampled Fourier transforms are particularly effective in (non-Bayesian) compressed sensing. Our work implies as similar conclusion in the Bayesian setting: indeed, substituting (4.2) and (4.3) into (3.5) we immediately deduce that the measurement condition for Bayesian recovery with $\mathcal{P}_s$ behaves like $m = \mathcal{O}(s^2)$ up to log terms.

**Remark 4.4 (Quadratic bottleneck)** Once more we witness a quadratic bottleneck. In the non-Bayesian setting, one can show stable and accurate recovery of sparse vectors from $\mathcal{O}(s)$ measurements, up to log terms (see, e.g., [5, Cor. 13.15]). However, this requires specialized theoretical techniques that heavily leverage the structure of the set of sparse vectors. In the setting of this paper, the bottleneck arises from the generality of the approach considered in this work: specifically, the fact that our main results hold for arbitrary probability distributions $\mathcal{P}$.

## 5 Limitations and future work

We end by discussing a number of limitations and avenues for future work. First, although our main result Theorem 3.1 is very general, we have only applied it to a number of different cases, such as Lipschitz pushforward measures and Gaussian random matrices or subsampled orthogonal transforms. We believe many other important problems can be studied as corollaries of our main results. This includes sampling with heavy-tailed vectors [50], sampling with random convolutions [67], multi-sensor acquisition problems [27], generative models augmented with sparse deviations [32], block sampling [3, 20, 24], with applications to practical MRI acquisition, sparse tomography [8], deconvolution and inverse source problems [9]. We believe our framework can also be applied to various types of non-Gaussian noise, as well as problems involving sparsely-corrupted measurements [50]. We are actively investigating applying our framework to these problems.

Second, as noted in Remarks 4.2 and 4.4 there is a quadratic bottleneck when considering subsampled orthogonal transforms. In the non-Bayesian case, this can be overcome in the case of (structured) sparse models using more technical arguments [3]. We believe similar ideas could also be exploited in the Bayesian setting. On a related note, both [16, 22] consider ReLU generative models in the non-Bayesian setting, and derive measurement conditions that do not involve the Lipschitz constant $L$ of the DNN. It is unclear whether analogous results can be established in the Bayesian setting.

Third, our main result involves the density shift bound (Definition 2.5). In particular, the noise distribution should have a density. This rather unpleasant technical assumption stems from Lemma C.6, which is a key step in proving the main result, Theorem 3.1. This lemma allows one to replace the 'real' distribution $\mathcal{R}$ in the probability term $p$ in Theorem 3.1 by the approximate distribution $\mathcal{P}$. This is done in order to align the prior and the posterior, which is necessary for the subsequent steps of the proof of Theorem 3.1. It would be interesting to see if this assumption on the noise could be removed through a refined analysis.

Fourth, as noted in [16], the coherence $\mu(U; D)$ arising in Theorem 3.9 may be high. In the non-Bayesian setting, this has been addressed in [2, 3, 17] by using a nonuniform probability distribution for drawing rows of $U$, with probabilities given in terms of so-called *local coherences* of $U$ with relative to $D$. As shown therein, this can lead to significant performance gains over sampling uniformly at random. We believe a similar approach can be considered in the Bayesian setting as a consequence of our general framework. We intend to explore this in future work.

Finally, our results in this paper are theoretical, and strive to study the sample complexity of Bayesian recovery. We do not address the practical problem of sampling from the posterior. This is a key computational challenge in Bayesian inverse problems [12]. However, efficient techniques for doing this are emerging. See, e.g., [48, 54, 70] and references therein. We believe an advantage of our results is their independence from the choice of posterior sampling algorithm, whose analysis can therefore be performed separately. This is an interesting problem we intend to examine in the future. In future work we also intend present numerical experiments for various practical settings that further support the theory developed in this paper.

## Acknowledgments and Disclosure of Funding

BA acknowledges the support of the Natural Sciences and Engineering Research Council of Canada of Canada (NSERC) through grant RGPIN-2021-611675. NH acknowledges support from an NSERC Canada Graduate Scholarship. Both authors would like to thank Paul Tupper and Weiran Sun for helpful comments and feedback.

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

# A  Covering number estimates and the proofs of Propositions 4.1 and 4.3

The proof of Proposition 4.1 relies on the following two lemmas.

**Lemma A.1** (Approximate covering number under a Lipschitz pushforward map)**.** *Let $G : \mathbb{R}^k \to \mathbb{R}^n$ be Lipschitz with constant $L \geq 0$, i.e.,*

$$\|G(x) - G(z)\| \leq L\|x - z\|, \quad \forall x, z \in \mathbb{R}^k,$$

*and define $\mathcal{P} = G\sharp\gamma$, where $\gamma$ is any probability distribution on $\mathbb{R}^k$. Then*

$$\mathrm{Cov}_{\eta,\delta}(\mathcal{P}) \leq \mathrm{Cov}_{\eta/L,\delta}(\gamma), \quad \forall \delta, \eta \geq 0.$$

*Proof.* Let $\{x_i\}_{i=1}^k \subseteq \mathrm{supp}(\gamma)$ and define $z_i = G(x_i) \in \mathrm{supp}(G\sharp\gamma)$ for $i = 1, \ldots, k$. Let $z \in G(B_{\eta/L}(x_i))$ and write $z = G(x)$ for some $x \in B_{\eta/L}(x_i)$. Then

$$\|z - z_i\| = \|G(x) - G(x_i)\| \leq L\|x - x_i\| \leq \eta.$$

Hence $z \in B_\eta(z_i)$. Since $z$ was arbitrary, we deduce that $G(B_{\eta/L}(x_i)) \subseteq B_\eta(z_i)$. It follows that $B_{\eta/L}(x_i) \subseteq G^{-1}(B_\eta(z_i))$. Now suppose that $\gamma\left[\bigcup_{i=1}^k B_{\eta/L}(x_i)\right] \geq 1 - \delta$. Then, by definition of the pushforward measure

$$G\sharp\gamma\left[\bigcup_{i=1}^k B_\eta(z_i)\right] = \gamma\left[\bigcup_{i=1}^k G^{-1}(B_\eta(z_i))\right] \geq \gamma\left[\bigcup_{i=1}^k B_{\eta/L}(x_i)\right] \geq 1 - \delta.$$

This gives the result. $\qquad\square$

**Lemma A.2** (Approximate covering number of a normal distribution)**.** *Let $\mathcal{P} = \mathcal{N}(0, \sigma^2 I)$ on $\mathbb{R}^n$. Then its approximate covering number (Definition 2.1) satisfies*

$$\mathrm{Cov}_{\eta,\delta}(\mathcal{P}) \leq \left(1 + \frac{2\sqrt{n}\sigma t}{\eta}\right)^n, \quad \text{where } t = 1 + \sqrt{\frac{2}{n}\log(1/\delta)}.$$

*Proof.* Observe that, for $t \geq 0$,

$$\mathcal{P}(B^c_{\sqrt{n}\sigma t}) = \mathbb{P}(X \geq nt^2) \leq (t^2 e^{1-t^2})^{n/2}.$$

where $X \sim \chi_n^2$ is a chi-squared random variable, and the inequality follows from a standard Chernoff bound. Now $t^2 \leq e^{2t}$ gives that

$$\mathcal{P}(B^c_{\sqrt{n}\sigma t}) \leq (e^{-(t-1)^2})^{n/2}.$$

Now set $t = 1 + \sqrt{\frac{2}{n}\log(1/\delta)}$ so that $\mathcal{P}(B^c_{\sqrt{n}\sigma t}) \leq \delta$. Hence, we have shown that

$$\mathrm{Cov}_{\eta,\delta}(\mathcal{P}) \leq \mathrm{Cov}_\eta(B_{\sqrt{n}\sigma t}),$$

where $\mathrm{Cov}_\eta$ is the classical covering number of a set, i.e.,

$$\mathrm{Cov}_\eta(A) = \min\left\{k : \exists\{x_i\}_{i=1}^k \subseteq A, A \subseteq \bigcup_{i=1}^k B_\eta(x_i)\right\}.$$

Using standard properties of covering numbers (see, e.g., [5, Lem. 13.22], we get

$$\mathrm{Cov}_\eta(B_{\sqrt{n}\sigma t}) = \mathrm{Cov}_{\eta/(\sqrt{n}\sigma t)}(B_1) \leq \left(1 + \frac{2\sqrt{n}\sigma t}{\eta}\right)^n,$$

as required. $\qquad\square$

*Proof of Proposition 4.1.* By Lemma A.1,

$$\mathrm{Cov}_{\eta,\delta}(\mathcal{P}) \leq \mathrm{Cov}_{\eta/L,\delta}(\mathcal{N}(0, I)).$$

The result now follows from Lemma A.2. $\qquad\square$

To prove Proposition 4.3, we first require the following lemma.

**Lemma A.3** (Approximate covering number of a mixture). *Let $\mathcal{P} = \sum_{i=1}^{r} p_i \mathcal{P}_i$ be a mixture of probability distributions $\mathcal{P}_i$ on $\mathbb{R}^n$, where $p_i \geq 0$, $\forall i$, and $\sum_{i=1}^{r} p_i = 1$. Then*

$$\mathrm{Cov}_{\eta,\delta}(\mathcal{P}) \leq \sum_{i=1}^{r} \mathrm{Cov}_{\eta,\delta}(\mathcal{P}_i).$$

*Proof.* For each $i = 1, \ldots, r$, let $\{x_j^{(i)}\}_{j=1}^{k_i} \subseteq \mathbb{R}^n$, and, in particular, $3x_j^{(i)} \in \mathrm{supp}(\mathcal{P}_i)$, be such that

$$\mathcal{P}_i \left( \bigcup_{j=1}^{k_i} B_\eta(x_j^{(i)}) \right) \geq 1 - \delta.$$

Then

$$\mathcal{P} \left( \bigcup_{i=1}^{r} \bigcup_{j=1}^{k_i} B_\eta(x_j^{(i)}) \right) = \sum_{i=1}^{r} p_i \mathcal{P}_i \left( \bigcup_{i=1}^{r} \bigcup_{j=1}^{k_i} B_\eta(x_j^{(i)}) \right)$$

$$\geq \sum_{i=1}^{r} p_i \mathcal{P}_i \left( \bigcup_{j=1}^{k_i} B_\eta(x_j^{(i)}) \right)$$

$$\geq \sum_{i=1}^{r} p_i (1 - \delta) = 1 - \delta.$$

Notice that $\mathrm{supp}(\mathcal{P}_i) \subseteq \mathrm{supp}(\mathcal{P})$, therefore, $x_j^{(i)} \in \mathrm{supp}(\mathcal{P})$. The result now follows. $\square$

*Proof of Proposition 4.3.* Let $\mathcal{S} = \{S : S \subseteq \{1, \ldots, n\}, |S| = s\}$. Then we can write $\mathcal{P}_s$ as the mixture

$$\mathcal{P}_s = \sum_{i=1}^{|\mathcal{S}|} \frac{1}{|\mathcal{S}|} \mathcal{P}_S,$$

where $\mathcal{P}_S$ is defined as follows: $x \sim \mathcal{P}_S$ if $x_i = 0$ for $i \notin S$ and, for $i \in S$, $x_i$ is drawn independently from the standard normal distribution on $\mathbb{R}$. Notice that $\mathcal{P}_S = G\sharp\gamma$, where $\gamma = \mathcal{N}(0, I)$ is the standard, multivariate normal distribution on $\mathbb{R}^s$ and $G : \mathbb{R}^s \to \mathbb{R}^n$ is a zero-padding map. The map $G$ is Lipschitz with constant $L = 1$. Hence, by Lemmas A.1 and A.2,

$$\mathrm{Cov}_{\eta,\delta}(\mathcal{P}_S) \leq \left( 1 + \frac{2\sqrt{s}t}{\eta} \right)^s, \quad t = 1 + \sqrt{\frac{2}{s} \log(1/\delta)}$$

We now apply Lemma A.3 and the fact that

$$|\mathcal{S}| = \binom{n}{s} \leq \left( \frac{en}{s} \right)^s,$$

the latter being a standard bound, to obtain

$$\mathrm{Cov}_{\eta,\delta}(\mathcal{P}_s) \leq \left( \frac{en}{s} \right)^s \left( 1 + \frac{2\sqrt{s}t}{\eta} \right)^s, \quad t = 1 + \sqrt{\frac{2}{s} \log(1/\delta)}.$$

Taking logarithms gives the result. $\square$

# B Concentration inequalities and the proofs of Theorems 3.5 and 3.9

We now aim to prove Theorems 3.5 and 3.9. To do this, we first derive concentration inequalities for subsgaussian random matrices and subsampled orthogonal transforms.

## B.1 Gaussian concentration and density shift bounds

**Lemma B.1** (Concentration and density shift bounds for Gaussian noise). *Let $\mathcal{E} = \mathcal{N}(0, \frac{\sigma^2}{m} I)$. Then the upper concentration bound $D_{\mathsf{upp}}(t; \mathcal{E})$ (Definition 2.4) can be taken as*

$$D_{\mathsf{upp}}(t; \mathcal{E}) = \left( \frac{t^2}{\sigma^2} e^{1 - \frac{t^2}{\sigma^2}} \right)^{m/2}, \quad \forall t > \sigma.$$

*and the density shift bound $D_{\mathsf{shift}}(\varepsilon, \tau; \mathcal{E})$ (Definition 2.5) can be taken as*

$$D_{\mathsf{shift}}(\varepsilon, \tau; \mathcal{E}) = \exp\left( \frac{m(2\tau + \varepsilon)}{2\sigma^2} \varepsilon \right), \quad \forall \varepsilon, \tau \geq 0.$$

*Proof.* Write $e \sim \mathcal{E}$ as $e = \frac{\sigma}{\sqrt{m}} n$, where $n \sim \mathcal{N}(0, I)$. Then

$$\mathbb{P}(\|e\| \geq t) = \mathbb{P}(\|n\|^2 \geq t^2 m/\sigma^2) = \mathbb{P}(X \geq t^2 m/\sigma^2),$$

where $X = \|n\|^2 \sim \chi_m^2$ is a chi-squared random variable with $m$ degrees of freedom. Using a standard Chernoff bound once more, we have

$$\mathbb{P}(X \geq zm) \leq (z\mathrm{e}^{1-z})^{m/2},$$

for any $z > 1$. Setting $z = \frac{t^2}{\sigma^2}$, we have

$$\mathbb{P}(\|e\| \geq t) \leq \left( \frac{t^2}{\sigma^2} e^{1 - \frac{t^2}{\sigma^2}} \right)^{m/2},$$

which gives the first result.

For the second result, we recall that $\mathcal{E}$ has density

$$p_{\mathcal{E}}(e) = (2\pi\sigma^2/m)^{-m/2} \exp\left( -\frac{m}{2\sigma^2} \|e\|^2 \right).$$

Therefore

$$\frac{p_{\mathcal{E}}(u)}{p_{\mathcal{E}}(v)} = \exp\left( \frac{m}{2\sigma^2} (\|v\| - \|u\|)(\|v\| + \|u\|) \right).$$

Now suppose that $\|u\| \leq \tau$ and $\|u - v\| \leq \varepsilon$. Then

$$\frac{p_{\mathcal{E}}(u)}{p_{\mathcal{E}}(v)} \leq \exp\left( \frac{m(2\tau + \varepsilon)}{2\sigma^2} \varepsilon \right).$$

Hence

$$D_{\mathsf{shift}}(\varepsilon, \tau; \mathcal{E}) \leq \exp\left( \frac{m(2\tau + \varepsilon)}{2\sigma^2} \varepsilon \right),$$

which gives the second result.

$\square$

## B.2 Subgaussian concentration inequalities

**Lemma B.2** (Lower and upper concentration bounds for subgaussian random matrices). *Let $\mathcal{A}$ be a distribution of subgaussian random matrices with parameters $\beta, \kappa > 0$ (Definition 3.4). Then the lower ad upper concentration bounds for $\mathcal{A}$ (Definition 2.3) can be taken as*

$$C_{\mathsf{upp}}(t; \mathcal{A}, \mathbb{R}^n) = C_{\mathsf{low}}(1/t; \mathcal{A}, \mathbb{R}^n) = 2\exp(-c(t, \beta, \kappa)m)$$

*for any $t > 1$, where $c(t, \beta, \kappa) > 0$ depends on $\beta, \kappa$ only.*

*Proof.* Let $x \in \mathbb{R}^n$ and observe that

$$\mathbb{P}(\|Ax\| \geq t\|x\|) \leq \mathbb{P}\left(\left|\|Ax\|^2 - \|x\|^2\right| \geq (t^2 - 1)\|x\|^2\right)$$
$$\mathbb{P}(\|Ax\| \leq t^{-1}\|x\|) \leq \mathbb{P}\left(\left|\|Ax\|^2 - \|x\|^2\right| \geq (1 - t^{-2})\|x\|^2\right). \tag{B.1}$$

We now use [39, Lem. 9.8]. Note that this result only considers a bound of the form $\mathbb{P}(|\|Ax\|^2 - \|x\|^2| \geq s\|x\|^2)$ for $s \in (0,1)$. But the proof straightforwardly extends to $s > 0$. □

## B.3 Concentration inequalities for randomly-subsampled orthogonal transforms

**Lemma B.3** (Concentration bounds for randomly-subsampled orthogonal transforms). *Let $D \subseteq \mathbb{R}^n$ and $\mathcal{A}$ be a distribution of randomly-subsampled orthognal transforms based on a matrix $U$ (Definition 3.7). Then the lower and upper concentration bounds for $\mathcal{A}$ (Definition 2.3) can be taken as*

$$C_{\mathsf{upp}}(t; \mathcal{A}, D) = C_{\mathsf{low}}(1/t; \mathcal{A}, D) = 2\exp\left(-\frac{mc(t)}{\mu(U; D)}\right)$$

*for any $t > 1$, where $\mu(U; \mathcal{D})$ is a in Definition 3.8 and $c(t) > 0$ depend on $t$ only.*

*Proof.* Due to (B.1), it suffices to bound

$$\mathbb{P}\left(\left|\|Ax\|^2 - \|x\|^2\right| \geq s\|x\|^2\right)$$

for $s > 0$. The result uses Bernstein's inequality for bounded random variables (see, e.g., [5, Thm. 12.18]). Let $x \in D$. By definition of $A$ and the fact that $U$ is orthogonal, we can write

$$\|Ax\|^2 - \|x\|^2 = \sum_{i=1}^{n} \left(\frac{n}{m}\mathbb{I}_{X_i=1} - 1\right)|\langle u_i, x\rangle|^2 =: \sum_{i=1}^{N} Z_i.$$

Notice that the random variables $Z_i$ are independent, with $\mathbb{E}(Z_i) = 0$. We also have

$$|Z_i| \leq \frac{n}{m}|\langle u_i, x\rangle|^2 \leq \frac{\mu(U; D)}{m}\|x\|^2 =: K$$

and

$$\sum_{i=1}^{n} \mathbb{E}|Z_i|^2 \leq \sum_{i=1}^{n} \frac{n^2|\langle u_i, x\rangle|^4}{m^2}\mathbb{E}(\mathbb{I}_{X_i=1}^2) \leq K\sum_{i=1}^{n} \frac{n|\langle u_i, x\rangle|^2}{m}\frac{m}{n} = K\|x\|^2 =: \sigma^2.$$

Therefore, by Bernstein's inequality,

$$\mathbb{P}(|\|Ax\|^2 - \|x\|^2| \geq s\|x\|^2) \leq 2\exp\left(-\frac{s^2\|x\|^4/2}{\sigma^2 + Ks\|x\|^2/3)}\right) = 2\exp\left(-\frac{ms^2/2}{\mu(U; D)(1 + s/3)}\right)$$

for any $s > 0$ and $x \in D$. The result now follows. □

**Lemma B.4** (Absolute concentration bounds for subsampled orthogonal transforms). *Let $D \subseteq \mathbb{R}^n$ and $\mathcal{A}$ be a distribution of randomly-subsampled orthogonal transforms based on a matrix $U$ (Definition 3.7). Then $\|A\| \leq \sqrt{n/m}$ a.s. for $A \sim \mathcal{A}$, and consequently the absolute concentration bound for $\mathcal{A}$ (Definition 2.3) can be taken as $C_{\mathsf{abs}}(s, t; \mathcal{A}, D) = 0$ for any $s \geq 0$ and $t \geq s\sqrt{n/m}$.*

*Proof.* Recall that $A$ consists of $q$ rows of an orthogonal matrix $U$ multiplied by the scalar $\sqrt{n/m}$. Hence $\|A\| \leq \sqrt{n/m}\|U\| = \sqrt{n/m}$. Now let $x \in D$ with $\|x\| \leq s$. Then $\|Ax\| \leq \|A\|\|x\| \leq \|A\|s$. Therefore $\|Ax\|\sqrt{n/ms} \leq t$, meaning that $\mathbb{P}(\|Ax\| > t) = 0$. This gives the result. □

## B.4 Proofs of Theorems 3.5 and 3.9

*Proof of Theorem 3.5.* Let $p = \mathbb{P}\left[\|x^* - \hat{x}\| \geq 34(\eta + \sigma)\right]$. We use Theorem 3.1 with $c = 32, c' = 2$, $t = 2$ and $\varepsilon$ replaced by $\varepsilon/d$, where $d \geq 1$ is a constant that will be chosen later. Let $\varepsilon' = \varepsilon/(d\delta^{1/p})$. Then Theorem 3.1 gives

$$
p \lesssim \delta + C_{\mathsf{abs}}(\varepsilon', 2\varepsilon'; \mathcal{A}, \mathbb{R}^n) + D_{\mathsf{upp}}(2\sigma; \mathcal{E})
$$
$$
+ 2D_{\mathsf{shift}}(2\varepsilon', 2\sigma; \mathcal{E})e^k \left[C_{\mathsf{low}}\left(1/2; \mathcal{A}, \mathbb{R}^n\right) + C_{\mathsf{upp}}\left(2; \mathcal{A}, \mathbb{R}^n\right) + 2D_{\mathsf{upp}}(2\sigma; \mathcal{E})\right].
$$

Consider $C_{\mathsf{abs}}(\varepsilon', 2\varepsilon'; \mathcal{A}, \mathbb{R}^n)$. If $x \in \mathbb{R}^n$ with $\|x\| \leq s$, then

$$
\mathbb{P}(\|Ax\| > t) \leq \mathbb{P}(\|Ax\| > (t/s)\|x\|).
$$

Hence, in this case, we may take

$$
C_{\mathsf{abs}}(\varepsilon', 2\varepsilon'; \mathcal{A}, \mathbb{R}^n) = C_{\mathsf{upp}}(2; \mathcal{A}, \mathbb{R}^n). \tag{B.2}
$$

Now by Lemma B.2, we have that

$$
C_{\mathsf{low}}(1/2; \mathcal{A}, \mathbb{R}^n) = C_{\mathsf{upp}}(2; \mathcal{A}, \mathbb{R}^n) = \exp(-c(\beta, \kappa)m),
$$

where $c(\beta, \kappa) > 0$ depends on $\beta, \kappa$ only. Also, by Lemma B.1, we have

$$
D_{\mathsf{upp}}(2\sigma; \mathcal{E}) = \left(2\mathrm{e}^{-1}\right)^{m/2} = \exp(-cm),
$$

for some universal constant $c > 0$ and

$$
D_{\mathsf{shift}}(2\varepsilon', 2\sigma; \mathcal{E}) = \exp\left(\frac{m(4\sigma + 2\varepsilon')}{2\sigma^2}2\varepsilon'\right) \leq \exp\left(\frac{6m}{d}\right),
$$

where we used the facts that $\sigma \geq \varepsilon/\delta^{1/p} = d\epsilon'$ and $d \geq 1$. We deduce that

$$
p \lesssim \delta + \exp(k + 6m/d - c(\beta, \kappa)m)
$$

for a possibly different constant $c(\beta, \kappa) > 0$. We now choose $d = d(\beta, \kappa) = 12/c(\beta, \kappa)$. Up to another possible change in $c(\beta, \kappa)$, the condition (3.4) on $m$ and (3.1) now give that $p \lesssim \delta$, as required. $\qquad\square$

*Proof of Theorem 3.9.* Let $p = \mathbb{P}\left[\|x^* - \hat{x}\| \geq 34(\eta + \sigma)\right]$. In this case, the forwards operator $A$ satisfies $\|A\| \leq \sqrt{n/m}$ (Lemma B.4). Hence we may apply Theorem 1.1 with $\theta = \sqrt{n/m}$ and $d = 2$ to obtain

$$
p \lesssim \delta + \mathrm{Cov}_{\eta, \delta}(\mathcal{P})\left[C_{\mathsf{low}}(1/2; \mathcal{A}, D) + C_{\mathsf{upp}}(2; \mathcal{A}, D) + \exp(-cm)\right]
$$

for some universal constant $c > 0$, where $D = \mathrm{supp}(\mathcal{P}) - \mathrm{supp}(\mathcal{P})$. Lemma B.3 now gives that

$$
p \lesssim \delta + \mathrm{Cov}_{\eta, \delta}(\mathcal{P})\left[\exp\left(-\frac{cm}{\mu(U; D)}\right) + \exp(-cm)\right]
$$

for a possibly different constant $c > 0$. The result now follows from the condition (3.5) on $m$. $\qquad\square$

# C Proofs of Theorems 1.1 and 3.1

We finally consider the proofs of the two general results, Theorems 1.1 and 3.1. Our main effort will be in establishing the latter, from which the former will follow after a short argument.

To prove Theorem 3.1, we first require some additional background on couplings, along with several lemmas. This is given in §C.1. We then establish a series of key technical lemmas, presented in §C.2, which are used in the main proof. Having shown these, the proof of Theorem 3.1 proceeds in §C.3 via a series of step. We now briefly describe these steps, and by doing so explain how the sets $D_1, D_2$ defined in (3.2)-(3.3) arise.

(i) First, using Lemma C.7, we decompose $\mathcal{P}, \mathcal{R}$ into distributions $\mathcal{P}', \mathcal{R}'$ that are supported in balls of a given radius, plus remainder terms. The distributions $\mathcal{P}', \mathcal{R}'$ are close (in $W_\infty$) to a discrete distribution $\mathcal{Q}$ supported at the centres of the balls that give the approximate cover satisfying (3.1).

(ii) Next, we replace $x^* \sim \mathcal{R}$ in the definition of the probability $p$ in Theorem 3.1 by $z^* \sim \mathcal{P}'$. The is done to align the prior with the posterior $\mathcal{P}(\cdot|y, A)$, which is needed later in the proof. We do this using Lemma C.6. Here we have to consider the action of $A$ on vectors of the form $x - z$, where $x \in \mathrm{supp}(\mathcal{R}')$ and $z \in \mathrm{supp}(\mathcal{P}')$. After determining the supports of $\mathcal{R}', \mathcal{P}'$, we see that $x - z \in D_1$, where $D_1$ is the set defined in (3.2).

(iii) We now decompose $\mathcal{P}'$ into a mixture over the balls mentioned in (i). After a series of arguments, we reduce the task to that of considering the probability that the conditional distribution is drawn from one ball when the prior is drawn from another. Lemmas C.4 and C.5 handle this. They involve estimating the action of $A$ on vectors $x - z$, where $z$ is the centre of one of the balls and $x$ comes from another ball. Since the balls are supported in $\mathrm{supp}(\mathcal{P})$ we have $x \in \mathrm{supp}(\mathcal{P})$, and since the centres come from the approximate covering number bound (3.1), we have that $z \in \mathrm{supp}(\mathcal{P})$ if $\mathcal{P}$ attains the minimum and $z \in \mathrm{supp}(\mathcal{R})$ otherwise. Hence $x - z \in D_2$, with $D_2$ as in (3.3).

## C.1 Background on couplings

For a number of our results, we require some background on couplings. We first recall some notation. Given probability spaces $(X, \mathcal{F}_1, \mu), (Y, \mathcal{F}_2, \nu)$, we write $\Gamma = \Gamma_{\mu, \nu}$ for the set of couplings, i.e., probability measures on the product space $(X \times Y, \sigma(\mathcal{F}_1 \otimes \mathcal{F}_2))$ whose marginals are $\mu$ and $\nu$, respectively. For convenience, we write $\pi_1 : X \times Y \to X$ and $\pi_2 : X \times Y \to Y$ for the projections $\pi_1(x, y) = x$ and likewise $\pi_2(x, y) = y$. In particular, for any coupling $\gamma$ we have $\pi_1 \sharp \gamma = \mu$ and $\pi_2 \sharp \gamma = \nu$, where $\sharp$ denotes the pushforward operation. As an immediate consequence, we observe that for any measurable function $\varphi : X \to \mathbb{R}$,

$$\int_X \varphi(x) \, d\mu(x) = \int_X \varphi(x) \, d\pi_1 \sharp \gamma(x) = \int_{X \times Y} \varphi(x) \, d\gamma(x, y). \tag{C.1}$$

Given a cost function $c : X \times Y \to [0, \infty)$, the Wasserstein-$p$ metric is defined as

$$W_p(\mu, \nu) = \inf_{\gamma \in \Gamma} \left( \int_{X \times Y} c(x, y)^p \, d\gamma(x, y) \right)^{1/p}$$

for $1 \leq p < \infty$ and

$$W_\infty(\mu, \nu) = \inf_{\gamma \in \Gamma} \left( \mathrm{esssup}_\gamma c(x, y) \right).$$

We say that $\gamma \in \Gamma$ is a $W_p$-*optimal coupling* of $\mu$ and $\nu$ if

$$\left( \int_{X \times Y} c(x, y)^p \, d\gamma(x, y) \right)^{1/p} = W_p(\mu, \nu)$$

for $1 \leq p < \infty$ or

$$\mathrm{esssup}_\gamma c(x, y) = W_\infty(\mu, \nu)$$

when $p = \infty$. Note that such a coupling exists whenever $X, Y$ are Polish spaces, and when the cost function is lower semicontinuous [41]. In our case, we generally work with Euclidean spaces with the cost function being the Euclidean norm, hence both conditions are satisfied.

For convenience, if $\gamma$ is probability measure on the product space $(X \times Y, \sigma(\mathcal{F}_1 \otimes \mathcal{F}_2))$, we will often write $\gamma(E_1, E_2)$ instead of $\gamma(E_1 \times E_2)$ for $E_i \in \mathcal{F}_i$, $i = 1, 2$. Moreover, if $x \in X$ is a singleton, we write $\gamma(x, E_2)$ for $\gamma(\{x\} \times E_2)$ and likewise for $\gamma(E_1, y)$.

We now need several lemmas on couplings.

**Lemma C.1.** *Suppose that $(X, \mathcal{F}_1, \mu), (Y, \mathcal{F}_2, \nu)$ are Borel probability spaces, and let $\gamma$ be a coupling of $\mu, \nu$ on the space $(X \times Y, \sigma(\mathcal{F}_1 \otimes \mathcal{F}_2))$. Then $\mathrm{supp}(\gamma) \subseteq \mathrm{supp}(\mu) \times \mathrm{supp}(\nu)$.*

*Proof.* Let $(x, y) \in \mathrm{supp}(\gamma)$. Then $\gamma(U_{x,y}) > 0$ for every open set $U_{x,y} \subseteq X \times Y$ that contains $(x, y)$. Now, to show that $(x, y) \in \mathrm{supp}(\mu) \times \mathrm{supp}(\nu)$, we show that $x \in \mathrm{supp}(\mu)$ and $y \in \mathrm{supp}(\nu)$.

Let $U_x \subseteq X$ be open with $x \in U_x$. By definition, $\mu(U_x) = \gamma(U_x \times \mathbb{R}^n)$. Since $U_x \times \mathbb{R}^n$ is open and contains $(x, y)$, it follows that $\gamma(U_x \times \mathbb{R}^n) > 0$. Since $U_x$ was arbitrary, we deduce that $x \in \mathrm{supp}(\mu)$. The argument that $y \in \mathrm{supp}(\nu)$ is identical. $\square$

**Lemma C.2.** *Let $X$ be a Polish space with a complete metric $d$. Let $\mu, \nu$ be Borel probability measures on $X$. Let $d_H$ be the Hausdorff metric with respect to $d$ and $W_\infty$ be the Wasserstein-$\infty$ metric with cost function $d$. Then*
$$d_H(\mathrm{supp}(\mu), \mathrm{supp}(\nu)) \leq W_\infty(\mu, \nu).$$
*In particular, $\mathrm{supp}(\mu) \subseteq B_\eta(\mathrm{supp}(\nu))$ for any $\eta \geq W_\infty(\mu, \nu)$.*

*Proof.* Since
$$d_H(\mathrm{supp}(\mu), \mathrm{supp}(\nu)) = \max \left\{ \sup_{x \in \mathrm{supp}(\nu)} d(x, \mathrm{supp}(\mu)), \sup_{y \in \mathrm{supp}(\mu)} d(y, \mathrm{supp}(\nu)) \right\}$$
we may, without loss of generality, assume that the maximum is achieved by $\sup_{x \in \mathrm{supp}(\nu)} d(x, \mathrm{supp}(\mu)) =: D$. Take a sequence $\{x_n\}_{n \in \mathbb{N}} \subseteq \mathrm{supp}(\nu)$ such that $D_n := d(x_n, \mathrm{supp}(\mu)) \to D$. Since $x_n \in \mathrm{supp}(\nu)$, for any $\varepsilon > 0$, we have $\nu(B_\varepsilon(x_n)) > 0$. Note that $B_\varepsilon(x_n)$ is measurable as we assume $X$ is Borel. For each $n \in \mathbb{N}$, define $\varepsilon_n = \frac{1}{n} D_n$. We show that for all $x \in B_{\varepsilon_n}(x_n), y \in \mathrm{supp}(\mu), d(x, y) > D_n(1 - \frac{1}{n})$. By triangle inequality, we have
$$d(x, y) \geq d(x_n, y) - d(x_n, x) \geq D_n - D_n/n = D_n(1 - 1/n).$$
Notice also that $D_n(1 - \frac{1}{n}) \leq D$ and converges to $D$ as $n \to \infty$. This implies that
$$A := \{(x', y') : d(x', y') > D_n(1 - 1/n)\} \supseteq B_{\varepsilon_n}(x_n) \times \mathrm{supp}(\mu).$$
Now consider any coupling $\gamma \in \Gamma_{\mu,\nu}$. We have
$$\gamma(A) \geq \gamma(B_{\varepsilon_n}(x_n) \times \mathrm{supp}(\mu)) = \gamma(B_{\varepsilon_n}(x_n) \times X) = \nu(B_{\varepsilon_n}(x_n)) > 0.$$
Therefore $\mathrm{ess\,sup}_\gamma d(x, y) > D_n(1 - 1/n)$. Now since $D_n(1 - \frac{1}{n}) \to D$ we have $\mathrm{ess\,sup}_\gamma d(x, y) \geq D$. This is holds for any coupling, therefore the result follows. $\square$

When working with a coupling between a finitely-supported distribution and a continuous distribution, the following lemma is often useful.

**Lemma C.3.** *Let $(X, \mathcal{F}_1, \mu), (Y, \mathcal{F}_2, \nu)$ be probability spaces, such that $\nu$ is finitely supported on a set $S \subseteq Y$. Let $\gamma$ be a coupling of $\mu, \nu$ and $E \subseteq X \times Y$ be $\gamma$-measurable. Write $E_y = \{x : (x, y) \in E\}$ for the slice of $E$ at $y \in Y$. Then*
$$\gamma(E) = \sum_{s \in S, s \in \pi_2(E)} \gamma(E_s \times \{s\}).$$

*Proof.* Write
$$\gamma(E) = \sum_{s \in S, s \in \pi_2(E)} \gamma(E_s \times \{s\}) + \gamma(\hat{E}),$$
where $\hat{E} = E \backslash \bigcup_{s \in S, s \in \pi_2(E)} (E_s \times \{s\})$. It suffices to show that $\gamma(\hat{E}) = 0$. Since $F \subseteq \pi_2^{-1}(\pi_2(F))$ for any set $F$, we have
$$\gamma(\hat{E}) \leq \gamma(\pi_2^{-1}(\pi_2(\hat{E})) = \nu(\pi_2(\hat{E})) = \nu(\pi_2(\hat{E}) \cap S).$$
But
$$\hat{E} = \{(x, y) \in E : y \notin S\}$$
and therefore $\pi_2(\hat{E}) \cap S = \emptyset$. The result now follows. $\square$

## C.2    Technical lemmas

### C.2.1    Separation lemma

The lemma considers a scenario where two random variables are drawn for a mixture of $k$ probability distributions. The second random variable is conditioned on the draw of the first. It then considers the probability that the two random variables are drawn from different distributions in the mixture, bounding this in terms of their Total Variation (TV) distance. It generalizes [49, Lem. 3.1].

**Lemma C.4** (Separation lemma). *Let $\mathcal{H}_1, \ldots, \mathcal{H}_k$ be Borel probability measures and consider the mixture $\mathcal{H} = \sum_{i=1}^k a_i \mathcal{H}_i$. Let $y^* \sim \mathcal{H}$ and $\hat{y} \sim \sum_{i=1}^k \mathbb{P}(y^* \sim \mathcal{H}_i | y^*) \mathcal{H}_i(\cdot | y^*)$ where $\mathbb{P}(y^* \sim \mathcal{H}_i | y^*)$ are the posterior weights. Then*

$$\mathbb{P}[y^* \sim \mathcal{H}_i, \hat{y} \sim \mathcal{H}_j(\cdot | y^*)] \leq 1 - \mathrm{TV}(\mathcal{H}_i, \mathcal{H}_j).$$

To clarify, in this lemma and elsewhere we use the notation $y^* \sim \mathcal{H}_i$ (and similar) to mean the event that $y^*$ is drawn from the $i$th distribution $\mathcal{H}_i$.

*Proof.* Note that if the $\mathcal{H}_i$ have densities $h_i$ with respect to some measure, these weights are given by

$$\mathbb{P}(y^* \sim \mathcal{H}_i | y^*) = \frac{a_i h_i(y^*)}{\sum_{j=1}^k a_j h_j(y^*)}. \tag{C.2}$$

We now write

$$
\begin{aligned}
p := \mathbb{P}[y^* \sim \mathcal{H}_i, \hat{y} \sim \mathcal{H}_j(\cdot | y^*)] &= \mathbb{P}[y^* \sim \mathcal{H}_i] \mathbb{P}[\hat{y} \sim \mathcal{H}_j(\cdot | y^*) | y^* \sim \mathcal{H}_j] \\
&= \mathbb{P}[y^* \sim \mathcal{H}_i] \mathbb{E}[\mathbb{P}(\hat{y} \sim \mathcal{H}_j(\cdot | y^*) | y^* \sim \mathcal{H}_i] \\
&= a_i \mathbb{E}[\mathbb{P}(\hat{y} \sim \mathcal{H}_j(\cdot | y^*)) | y^* \sim \mathcal{H}_i].
\end{aligned}
$$

Since $\mathbb{E}[y^* | y^* \sim \mathcal{H}_i] \sim \mathcal{H}_i$, we have

$$p = a_i \mathbb{E}[\mathbb{P}(\hat{y} \sim \mathcal{H}_j(\cdot | y^*)) | y^* \sim \mathcal{H}_i] = a_i \int \mathbb{P}(\hat{y} \sim \mathcal{H}_j(\cdot | y^*)) \, d\mathcal{H}_i(y^*).$$

Now, because of the mixture property, $\mathcal{H}_i \ll \mathcal{H}$ and therefore its Radon-Nikodym derivative $h_i = \frac{d\mathcal{H}_i}{d\mathcal{H}}$ exists. This means we may write

$$p = a_i \int \mathbb{P}(\hat{y} \sim \mathcal{H}_j(\cdot | y^*)) h_i(y^*) \, d\mathcal{H}(y^*).$$

By definition, we have $\mathbb{P}(\hat{y} \sim \mathcal{H}_j(\cdot | y^*)) = \mathbb{P}(y^* \sim \mathcal{H}_j | y^*)$ and using (C.2), we deduce that

$$p = \int \frac{a_i a_j h_i(y^*) h_j(y^*)}{\sum_{l=1}^k a_l h_l(y^*)} \, d\mathcal{H}(y^*)$$

We now write

$$
\begin{aligned}
p &= \int \frac{a_i h_i(y^*) h_j(y^*) a_j}{a_i h_i(y^*) + a_j h_j(y^*) + \sum_{l \neq i,j} a_l h_l(y^*)} \, d\mathcal{H}(y^*) \\
&\leq \int \frac{a_i h_i(y^*) h_j(y^*) a_j}{a_i h_i(y^*) + a_j h_j(y^*)} \, d\mathcal{H}(y^*) \\
&= \int \frac{a_i h_i(y^*) h_j(y^*) a_j}{(a_i h_i(y^*) + a_j h_j(y^*))} \, d\mathcal{H}(y^*) \\
&= \int \frac{a_i h_i(y^*) h_j(y^*) a_j}{a_i h_i(y^*) + a_j h_j(y^*)} \, d\mathcal{H}(y^*) \\
&\leq \int \frac{a_i h_i(y^*) h_j(y^*) a_j}{\max\{a_i h_i(y^*), a_j h_j(y^*)\}} \, d\mathcal{H}(y^*) \\
&= \int \min\{a_i h_i(y^*), a_j h_j(y^*)\} \, d\mathcal{H}(y^*) \\
&= 1 - \int \frac{1}{2}(h_i(y^*) + h_j(y^*)) - \min\{a_i h_i(y^*), a_j h_j(y^*)\} \, d\mathcal{H}(y^*) \\
&= 1 - \int \frac{1}{2}|h_i(y^*) - h_j(y^*)| \, d\mathcal{H}(y^*) \\
&= 1 - \mathrm{TV}(\mathcal{H}_i, \mathcal{H}_j),
\end{aligned}
$$

as required. □

### C.2.2 Disjointly-supported measures induce well-separated measurement distributions

The following lemma pertains to the pushforwards of measures supported in $\mathbb{R}^n$ via the forward operator $A$ and noise $e$. Specifically, it states that if two distributions $\mathcal{P}_{\text{int}}$ and $\mathcal{P}_{\text{ext}}$ are disjointly supported then their corresponding pushforwards $\mathcal{H}_{\text{int},A}$ and $\mathcal{H}_{\text{ext},A}$ are, on average with respect to $A \sim \mathcal{A}$, well-separated, in the sense of their TV-distance. It is generalization of [49, Lem. 3.2] that allows for arbitrary distributions $\mathcal{A}$ of the forward operators, as opposed to just distributions of Gaussian random matrices.

**Lemma C.5** (Disjointly-supported measures induce well-separated measurement distributions). *Let $\tilde{x} \in \mathbb{R}^n, \sigma \geq 0, \eta \geq 0, c \geq 1$, $\mathcal{P}_{\text{ext}}$ be a distribution supported in the set*

$$S_{\tilde{x},\text{ext}} = \{x \in \mathbb{R}^n : \|x - \tilde{x}\| \geq c(\eta + \sigma)\}$$

*and $\mathcal{P}_{\text{int}}$ be a distribution supported in the set*

$$S_{\tilde{x},\text{int}} = \{x \in \mathbb{R}^n : \|x - \tilde{x}\| \leq \eta\}.$$

*Given $A \in \mathbb{R}^{m \times n}$, let $\mathcal{H}_{\text{int},A}$ be the distribution of $y = Ax^* + e$ where $x^* \sim \mathcal{P}_{\text{int}}$ and $e \sim \mathcal{E}$ independently, and define $\mathcal{H}_{\text{ext},A}$ in a similar way. Then*

$$\mathbb{E}_{A \sim \mathcal{A}}[\text{TV}(\mathcal{H}_{\text{int},A}, \mathcal{H}_{\text{ext},A})] \geq 1 - \left[ C_{\text{low}}\left(\frac{2}{\sqrt{c}}; \mathcal{A}, D_{\text{ext}}\right) + C_{\text{upp}}\left(\frac{\sqrt{c}}{2}; \mathcal{A}, D_{\text{int}}\right) + 2D_{\text{upp}}\left(\frac{\sqrt{c}\sigma}{2}; \mathcal{E}\right) \right],$$

*where $D_{\text{ext}} = \{x - \tilde{x} : x \in \text{supp}(\mathcal{P}_{\text{ext}})\}$, $D_{\text{int}} = \{x - \tilde{x} : x \in \text{supp}(\mathcal{P}_{\text{int}})\}$ and $C_{\text{upp}}(\cdot; \mathcal{A})$, $C_{\text{low}}(\cdot; \mathcal{A})$ and $D_{\text{upp}}(\cdot; \mathcal{E})$ are as in Definitions 2.3 and 2.4, respectively.*

Notice that the average TV-distance is bounded below by the concentration bounds $C_{\text{low}}$ and $C_{\text{upp}}$ for $\mathcal{A}$ (Definition 2.3) and the concentration bound $D_{\text{upp}}$ for $\mathcal{E}$ (Definition 2.4). This is unsurprising. The pushforward measures are expected to be well-separated if, firstly, the action of $A$ approximately preserves the lengths of vectors (which explains the appearance of $C_{\text{low}}$ and $C_{\text{upp}}$) and, secondly, adding noise by $\mathcal{E}$ does not, with high probability, cause well-separated vectors to become close to each other (which explains the appearance of $D_{\text{upp}}$). Also as expected, as $c$ increases, i.e., the distributions $\mathcal{P}_{\text{int}}$ and $\mathcal{P}_{\text{ext}}$ become further separated, the average TV-distance increases.

*Proof.* Given $A \in \mathbb{R}^{m \times n}$, let

$$B_A = \{y \in \mathbb{R}^m : \|y - A\tilde{x}\| \leq \sqrt{c}(\eta + \sigma)\}.$$

We claim that

$$\mathbb{E}_A[\mathcal{H}_{\text{ext},A}(B_A)] \leq C_{\text{low}}\left(\frac{2}{\sqrt{c}}; \mathcal{A}, D_{\text{ext}}\right) + D_{\text{upp}}(\sigma\sqrt{c}; \mathcal{E}), \tag{C.3}$$

$$\mathbb{E}_A[\mathcal{H}_{\text{int},A}(B_A)] \geq 1 - \left[ C_{\text{upp}}\left(\frac{\sqrt{c}}{2}; \mathcal{A}, D_{\text{int}}\right) + D_{\text{upp}}\left(\frac{\sqrt{c}}{2}\sigma; \mathcal{E}\right) \right]. \tag{C.4}$$

Notice that these claims immediately imply the result, since

$$\mathbb{E}_{A \sim \mathcal{A}}\text{TV}(\mathcal{H}_{\text{ext},A}, \mathcal{H}_{\text{int},A}) \geq \mathbb{E}_{A \sim \mathcal{A}}[\mathcal{H}_{\text{int},A}(B_A)] - \mathbb{E}_{A \sim \mathcal{A}}[\mathcal{H}_{\text{ext},A}(B_A)].$$

Therefore, the rest of the proof is devoted to showing (C.3) and (C.4). For the former, we write

$$\mathbb{E}_{A \sim \mathcal{A}}[\mathcal{H}_{\text{ext},A}(B_A)] = \mathbb{E}_{A \sim \mathcal{A}}\left[ \int \int 1_{B_A}(Ax + e)\, d\mathcal{P}_{\text{ext}}(x)\, d\mathcal{E}(e) \right]$$

$$= \mathbb{E}_{A \sim \mathcal{A}}\left[ \int \int 1_{B_A}(Ax + e)\, d\mathcal{E}(e)\, d\mathcal{P}_{\text{ext}}(x) \right] \tag{C.5}$$

$$= \mathbb{E}_{A \sim \mathcal{A}}[\mathbb{E}_{x \sim \mathcal{P}_{\text{ext}}}[\mathcal{E}(B_A - Ax)]]$$

$$= \mathbb{E}_{x \sim \mathcal{P}_{\text{ext}}}[\mathbb{E}_{A \sim \mathcal{A}}[\mathcal{E}(B_A - Ax)]],$$

where $B_A - Ax = \{b - Ax : b \in B_A\}$. We now bound $E_{x \sim \mathcal{P}_{\text{ext}}}[\mathbb{E}_{A \sim \mathcal{A}}\mathcal{E}(B_A - Ax)]$. Given $x \in \mathbb{R}^n$, let $C_x = \{A : \|Ax - A\tilde{x}\| < 2\sqrt{c}(\eta + \sigma)\} \subseteq \mathbb{R}^{m \times n}$ and write

$$I_1 = \mathbb{E}_{x \sim \mathcal{P}_{\text{ext}}}[\mathbb{E}_{A \sim \mathcal{A}}\mathcal{E}(B_A - Ax)1_{C_x}], \quad I_2 = \mathbb{E}_{x \sim \mathcal{P}_{\text{ext}}}[\mathbb{E}_{A \sim \mathcal{A}}\mathcal{E}(B_A - Ax)1_{C_x^c}]$$

so that
$$\mathbb{E}_{x\sim\mathcal{P}_{\text{ext}}}[\mathbb{E}_{A\sim\mathcal{A}}[\mathcal{E}(B_A - Ax)]] = I_1 + I_2. \tag{C.6}$$

We will bound $I_1, I_2$ separately. For $I_1$, we first write

$$\begin{aligned}
I_1 &= \mathbb{E}_{x\sim\mathcal{P}_{\text{ext}}}[\mathbb{E}_{A\sim\mathcal{A}}[\mathcal{E}(B_A - Ax)1_{C_x}]] \\
&\leq \mathbb{E}_{x\sim\mathcal{P}_{\text{ext}}}[\mathbb{E}_{A\sim\mathcal{A}}[1_{C_x}]] \\
&= \mathbb{E}_{x\sim\mathcal{P}_{\text{ext}}}[\mathbb{P}_{A\sim\mathcal{A}}(\|Ax - A\tilde{x}\| < 2\sqrt{c}(\eta + \sigma))],
\end{aligned}$$

where the inequality follows from the fact that $\mathcal{E}(B_A - Ax) \leq 1$. Now since $x \sim \mathcal{P}_{\text{ext}}$, we have $x \in S_{\tilde{x},\text{ext}}$ and therefore $\|x - \tilde{x}\| \geq c(\eta + \sigma)$. Hence

$$\mathbb{E}_{x\sim\mathcal{P}_{\text{ext}}}[\mathbb{P}_{A\sim\mathcal{A}}(\|Ax - A\tilde{x}\| < 2\sqrt{c}(\eta + \sigma)] \leq \mathbb{E}_{x\sim\mathcal{P}_{\text{ext}}}\left[\mathbb{P}_{A\sim\mathcal{A}}\left(\|Ax - A\tilde{x}\| < \frac{2}{\sqrt{c}}\|x - \tilde{x}\|\right)\right].$$

Since the outer expectation term has $x \sim \mathcal{P}_{\text{ext}}$, we have that $x \in \text{supp}(\mathcal{P}_{\text{ext}})$ with probability one. Using Definition 2.3, we deduce that

$$I_1 \leq C_{\text{low}}\left(\frac{2}{\sqrt{c}}; \mathcal{A}, D_{\text{ext}}\right). \tag{C.7}$$

We now bound $I_2$. Let $x \in S_{\tilde{x},\text{ext}}$ and $A \in C_x^c$, i.e., $\|A(x - \tilde{x})\| > 2\sqrt{c}(\eta + \sigma)$. We now show that $B_A \subseteq B_{A,x}$, where $B_{A,x} = \{y \in \mathbb{R}^m : \|y - Ax\| \geq \sqrt{c}(\eta + \sigma)\}$. Suppose that $y \in B_A$, i.e., $\|y - A\tilde{x}\| \leq \sqrt{c}(\eta + \sigma)$. We have

$$\begin{aligned}
\|y - Ax\| &= \|y - A\tilde{x} + A\tilde{x} - Ax\| \\
&\geq \|A(\tilde{x} - x)\| - \|y - A\tilde{x}\| \\
&> 2\sqrt{c}(\eta + \sigma) - \sqrt{c}(\eta + \sigma) \\
&= \sqrt{c}(\eta + \sigma),
\end{aligned}$$

and therefore $y \in B_{A,x}$, as required. Using this, we have

$$\mathcal{E}(B_A - Ax) \leq \mathcal{E}(B_{A,x} - Ax), \quad \forall A \in C_x^c, \ x \in S_{\tilde{x},\text{ext}},$$

and therefore

$$I_2 = \mathbb{E}_{x\sim\mathcal{P}_{\text{ext}}}[\mathbb{E}_{A\sim\mathcal{A}}[\mathcal{E}(B_A - Ax)1_{C_x^c}]] \leq \mathbb{E}_{x\sim\mathcal{P}_{\text{ext}}}[\mathbb{E}_{A\sim\mathcal{A}}[\mathcal{E}(B_{A,x} - Ax)1_{C_x^c}]].$$

But we notice that $B_{A,x} - Ax = B_{\sqrt{c}(\eta+\sigma)}^c$. Now since $\eta \geq 0$, we have $B_{\sqrt{c}(\eta+\sigma)}^c \subseteq B_{\sigma\sqrt{c}}^c$. Hence

$$\mathcal{E}(B_{A,x} - Ax) = \mathcal{E}(B_{\sqrt{c}(\eta+\sigma)}^c) \leq \mathcal{E}(B_{\sigma\sqrt{c}}^c) \leq D_{\text{upp}}(\sigma\sqrt{c}; \mathcal{E}),$$

and therefore $I_2 \leq D_{\text{upp}}(\sigma\sqrt{c}; \mathcal{E})$. Combining this with (C.5), (C.6) and (C.7), we deduce that

$$\mathbb{E}_A[\mathcal{H}_{\text{ext},A}(B_A)] \leq \mathbb{E}_{x\sim\mathcal{P}_{\text{ext}}}[\mathbb{E}_{A\sim\mathcal{A}}[\mathcal{E}(B_A - Ax)]] = I_1 + I_2 \leq C_{\text{low}}(2/\sqrt{c}; \mathcal{A}, D_{\text{ext}}) + C_{\text{upp}}(\sigma\sqrt{c}; \mathcal{E}),$$

which shows (C.3).

We will now establish (C.4). With similar reasoning to (C.5), we have

$$\mathbb{E}_{A\sim\mathcal{A}}[\mathcal{H}_{\text{int},A}(B_A^c)] = \mathbb{E}_{x\sim\mathcal{P}_{\text{int}}}[\mathbb{E}_{A\sim\mathcal{A}}[\mathcal{E}(B_A^c - Ax)]]$$

Proceeding as before, let $D_x = \{A : \|Ax - A\tilde{x}\| < \frac{\sqrt{c}}{2}(\eta + \sigma)\}$, $I_1 = \mathbb{E}_{x\sim\mathcal{P}_{\text{int}}}[\mathbb{E}_{A\sim\mathcal{A}}[\mathcal{E}(B_A^c - Ax)]1_{D_x^c}]$, and $I_2 = \mathbb{E}_{x\sim\mathcal{P}_{\text{int}}}[\mathbb{E}_{A\sim\mathcal{A}}[\mathcal{E}(B_A^c - Ax)]1_{D_x}]$ so that

$$\mathbb{E}_{A\sim\mathcal{A}}[\mathcal{H}_{\text{int},A}(B_A^c)] = \mathbb{E}_{x\sim\mathcal{P}_{\text{int}}}[\mathbb{E}_{A\sim\mathcal{A}}[\mathcal{E}(B_A^c - Ax)]] = I_1 + I_2. \tag{C.8}$$

The terms $I_1, I_2$ are similar to those considered in the previous case. We bound them similarly. For $I_1$, we have, by dropping the inner probability terms,

$$I_1 \leq \mathbb{E}_{x\sim\mathcal{P}_{\text{int}}}[\mathbb{E}_{A\sim\mathcal{A}}[1_{D_x^c}]] = \mathbb{E}_{x\sim\mathcal{P}_{\text{int}}}\left[\mathbb{P}_{A\sim\mathcal{A}}(\|A(x - \tilde{x})\| \geq \frac{\sqrt{c}}{2}(\eta + \sigma))\right].$$

Since $x \in S_{\tilde{x},\text{int}}$, we have $\|x - \tilde{x}\| \leq \eta \leq \eta + \sigma$ which gives

$$\mathbb{E}_{x\sim\mathcal{P}_{\text{int}}}[\mathbb{P}_{A\sim\mathcal{A}}(\|A(x - \tilde{x})\| \geq \frac{\sqrt{c}}{2}(\eta + \sigma))] \leq \mathbb{E}_{x\sim\mathcal{P}_{\text{int}}}\left[\mathbb{P}_{A\sim\mathcal{A}}(\|A(x - \tilde{x})\| \geq \frac{\sqrt{c}}{2}\|x - \tilde{x}\|)\right]$$

and therefore

$$I_1 \leq C_{\mathsf{upp}} \left( \frac{\sqrt{c}}{2}; \mathcal{A}, D_{\mathsf{int}} \right). \tag{C.9}$$

We now bound $I_2$. Let $x \in S_{\tilde{x},\mathsf{int}}$ and suppose that $A \in D_x$, i.e., $\|x - \tilde{x}\| \leq \eta$ and $\|A(x - \tilde{x})\| < \frac{\sqrt{c}}{2}(\eta + \sigma)$. Define $\hat{B}_{A,x} = \{y \in \mathbb{R}^m : \|y - Ax\| < \frac{\sqrt{c}}{2}(\eta + \sigma)\}$. We will show $\hat{B}_{A,x} \subseteq B_A$ in this case. Let $y \in B_{A,x}$. Then

$$\|y - A\tilde{x}\| \leq \|y - Ax\| + \|Ax - A\tilde{x}\| < \frac{\sqrt{c}}{2}(\eta + \sigma) + \frac{\sqrt{c}}{2}(\eta + \sigma) = \sqrt{c}(\eta + \sigma),$$

as required. This implies that $B_A^c \subseteq \hat{B}_{A,x}^c$. Hence

$$\mathcal{E}(B_A^c - Ax) \leq \mathcal{E}(\hat{B}_{A,x}^c - Ax) = \mathcal{E}(B_{\frac{\sqrt{c}}{2}(\eta+\sigma)}^c) \leq \mathcal{E}(B_{\frac{\sqrt{c}}{2}\sigma}^c)$$

which implies that $I_2 \leq D_{\mathsf{upp}}(\frac{\sqrt{c}}{2}\sigma; \mathcal{E})$. Combining with (C.8) and (C.9) we get

$$\mathbb{E}_A[\mathcal{H}_{\mathsf{int},A}(B_A^c)] \leq \mathbb{E}_{x \sim \mathcal{P}_{\mathsf{int}}}[\mathbb{E}_{A \sim \mathcal{A}}[\mathcal{E}((B_A - Ax)^c)]] \leq C_{\mathsf{upp}}\left(\frac{\sqrt{c}}{2}; \mathcal{A}, D_{\mathsf{int}}\right) + D_{\mathsf{upp}}\left(\frac{\sqrt{c}}{2}\sigma; \mathcal{E}\right),$$

which implies (C.4). This completes the proof. $\qquad\square$

### C.2.3 Replacing the real distribution with the approximate distribution

We next establish a result that allows one to upper bound the failure probability based on draws from the real distribution $\mathcal{R}$ with the failure probability based on draws from the approximate distribution $\mathcal{P}$. This lemma is a key technical step that aligns the prior distribution with the posterior. The specific bound is given in terms of the Wasserstein distance between $\mathcal{R}$ and $\mathcal{P}$ and several of the concentration bounds defined in §2. This is a significant generalization of [49, Lem. 3.3] that allows for arbitrary distributions $\mathcal{A}, \mathcal{E}$ for the forwards operator and noise.

**Lemma C.6** (Replacing the real distribution with the approximate distribution). *Let $\varepsilon, \sigma, d, t \geq 0$, $c \geq 1$, $\mathcal{E}$ be a distribution on $\mathbb{R}^m$ and $\mathcal{R}, \mathcal{P}$ be distributions on $\mathbb{R}^n$ such that $W_\infty(\mathcal{R}, \mathcal{P}) \leq \varepsilon$. Let $\Pi$ be an $W_\infty$-optimal coupling of $\mathcal{R}$ and $\mathcal{P}$ and define the set $D = \{x^* - z^* : (x^*, z^*) \in \mathrm{supp}(\Pi)\}$. Let*

$$p = \mathbb{P}_{x^* \sim \mathcal{R}, A \sim \mathcal{A}, e \sim \mathcal{E}, \hat{x} \sim \mathcal{P}(\cdot | Ax^* + e, A)}[\|x^* - \hat{x}\| \geq d + \varepsilon]$$

*and*

$$q = \mathbb{P}_{z^* \sim \mathcal{P}, A \sim \mathcal{A}, e \sim \mathcal{E}, \hat{z} \sim \mathcal{P}(\cdot | Az^* + e, A)}[\|z^* - \hat{z}\| \geq d].$$

*Then*

$$p \leq C_{\mathsf{abs}}(\varepsilon, t\varepsilon; \mathcal{A}, D) + D_{\mathsf{upp}}(c\sigma; \mathcal{E}) + D_{\mathsf{shift}}(t\varepsilon, c\sigma; \mathcal{E})q,$$

*where $C_{\mathsf{abs}}(\varepsilon, t\varepsilon; \mathcal{A}, D)$, $D_{\mathsf{upp}}(c\sigma; \mathcal{E})$ and $D_{\mathsf{shift}}(t\varepsilon, c\sigma; \mathcal{E})$ are as in Definitions 2.3, 2.4 and 2.5, respectively.*

As expected, this lemma involves a trade-off. The constant $C_{\mathsf{abs}}(\varepsilon, t\varepsilon; \mathcal{A}, D)$ is made smaller (for fixed $\varepsilon$) by making the constant $t$ larger. However, this increases $D_{\mathsf{shift}}(t\varepsilon, c\sigma; \mathcal{E})$, which is compensated by making $c$ smaller. However, this in turn increases the constant $D_{\mathsf{upp}}(c\sigma; \mathcal{E})$.

*Proof.* Define the events

$$B_{1,\hat{x}} = \{x^* : \|x^* - \hat{x}\| \geq d + \varepsilon\}, \quad B_{2,\hat{z}} = \{z^* : \|z^* - \hat{z}\| \geq d\}$$

so that

$$\begin{aligned} p &= \mathbb{P}_{x^* \sim \mathcal{R}, A \sim \mathcal{A}, e \sim \mathcal{E}, \hat{x} \sim \mathcal{P}(\cdot | Ax^* + e, A)}[x^* \in B_{1,\hat{x}}] \\ q &= \mathbb{P}_{z^* \sim \mathcal{P}, A \sim \mathcal{A}, e \sim \mathcal{E}, \hat{z} \sim \mathcal{P}(\cdot | Az^* + e, A)}[z^* \in B_{2,\hat{z}}] \end{aligned}. \tag{C.10}$$

Observe that

$$\begin{aligned} p &= \mathbb{E}_{x^* \sim \mathcal{R}}[\mathbb{E}_{A \sim \mathcal{A}}\mathbb{E}_{y|A,x^*}[\mathbb{E}_{\hat{x} \sim \mathcal{P}(\cdot | Ax^* + e, A)}[1_{B_{1,\hat{x}}}]]] \\ &= \int\int\int\int 1_{B_{1,\hat{x}}}(x^*) d\mathcal{P}(\cdot | Ax^* + e, A)(\hat{x}) d\mathcal{E}(e) d\mathcal{A}(A) d\mathcal{R}(x^*) \end{aligned}$$

and similarly

$$q = \int \int \int \int 1_{B_{2,\hat{z}}}(z^*) d\mathcal{P}(\cdot|Az^* + e, A)(\hat{z}) d\mathcal{E}(e) d\mathcal{A}(A) d\mathcal{P}(z^*).$$

Therefore, to obtain the result, it suffices to replace samples from the real distribution $\mathcal{R}$ with samples from the approximate distribution $\mathcal{P}$ and to replace the indicator function of $B_{1,\hat{x}}$ by the indicator function over $B_{2,\hat{z}}$. For the first task, we use couplings. Since $W_\infty(\mathcal{R}, \mathcal{P}) \leq \varepsilon$, there exists a coupling $\Pi$ between $\mathcal{R}, \mathcal{P}$ with $\Pi(\|x^* - z^*\| \leq \varepsilon) = 1$. By (C.1), we can write

$$p = \int \int \int \int 1_{B_{1,\hat{x}}}(x^*) d\mathcal{P}(\cdot|Ax^* + e, A)(\hat{x}) d\mathcal{E}(e) d\mathcal{A}(A) d\Pi(x^*, z^*).$$

Define $E = \{(x^*, z^*) : \|x^* - z^*\| \leq \varepsilon\}$ and observe that $\Pi(E) = 1$. Then, for fixed $A$, $e$, we have

$$\int \int 1_{B_{1,\hat{x}}}(x^*) d\mathcal{P}(\cdot|Ax^* + e, A)(\hat{x}) d\Pi(x^*, z^*) = \int_E \int 1_{B_{1,\hat{x}}}(x^*) d\mathcal{P}(\cdot|Ax^* + e, A)(\hat{x}) d\Pi(x^*, z^*)$$

$$\int \int 1_{B_{2,\hat{x}}}(z^*) d\mathcal{P}(\cdot|Ax^* + e, A)(\hat{x}) d\Pi(x^*, z^*) = \int_E \int 1_{B_{2,\hat{x}}}(z^*) d\mathcal{P}(\cdot|Ax^* + e, A)(\hat{x}) d\Pi(x^*, z^*).$$

We now show $1_{B_{1,\hat{x}}}(x^*) \leq 1_{B_{2,\hat{x}}}(z^*)$ for $(x^*, z^*) \in E$. Let $(x^*, z^*) \in E$ and suppose that $x^* \in B_{1,\hat{x}}$. Then $\|x^* - \hat{x}\| \geq d + \varepsilon$ and, since $\|x^* - z^*\| \leq \varepsilon$, we also have that $\|z^* - \hat{x}\| \geq d$ and therefore $z^* \in B_{2,\hat{x}}$, as required. Hence

$$\int 1_{B_{1,\hat{x}}}(x^*) d\mathcal{P}(\cdot|Ax^* + e, A)(\hat{x}) \leq \int 1_{B_{2,\hat{x}}}(z^*) d\mathcal{P}(\cdot|Ax^* + e, A)(\hat{x})$$

for $(x^*, z^*) \in E$. Now, since indicator functions are non-negative, Fubini's theorem immediately implies that

$$p = \int \int \int \int 1_{B_{1,\hat{x}}}(x^*) \, d\mathcal{P}(\cdot|Ax^* + e, A)(\hat{x}) \, d\mathcal{E}(e) \, d\mathcal{A}(A) \, d\Pi(x^*, z^*)$$

$$\leq \int \int \int \int 1_{B_{2,\hat{x}}}(z^*) \, d\mathcal{P}(\cdot|Ax^* + e, A)(\hat{x}) \, d\mathcal{E}(e) \, d\mathcal{A}(A) \, d\Pi(x^*, z^*).$$

Having introduced the coupling $\Pi$ and replaced $1_{B_{1,\hat{x}}}$ by $1_{B_{2,\hat{x}}}$, to establish the result it remains to replace the conditional distribution $\mathcal{P}(\cdot|Ax^* + e, A)$ by $\mathcal{P}(\cdot|Az^* + e, A)$. With a similar technique to that used in the proof of Lemma C.5, we define $C_{x^*, z^*} = \{A : \|A(x^* - z^*)\| > t\varepsilon\}$ and

$$I_1 = \int \int 1_{C_{x^*, z^*}}(A) \int \int 1_{B_{2,\hat{x}}}(z^*) \, d\mathcal{P}(\cdot|Ax^* + e, A)(\hat{x}) \, d\mathcal{E}(e) \, d\mathcal{A}(A) \, d\Pi(x^*, z^*)$$

$$I_2 = \int \int 1_{C^c_{x^*, z^*}}(A) \int \int 1_{B_{2,\hat{x}}}(z^*) \, d\mathcal{P}(\cdot|Ax^* + e, A)(\hat{x}) \, d\mathcal{E}(e) \, d\mathcal{A}(A) \, d\Pi(x^*, z^*)$$

so that

$$p \leq \int \int \int \int 1_{B_{2,\hat{x}}}(z^*) \, d\mathcal{P}(\cdot|Ax^* + e, A)(\hat{x}) \, d\mathcal{E}(e) \, d\mathcal{A}(A) \, d\Pi(x^*, z^*) = I_1 + I_2. \quad \text{(C.11)}$$

We first bound $I_1$. As before, we write

$$I_1 = \int \int 1_{C_{x^*, z^*}}(A) \int \int 1_{B_{2,\hat{x}}}(z^*) \, d\mathcal{P}(\cdot|Ax^* + e, A)(\hat{x}) \, d\mathcal{E}(e) \, d\mathcal{A}(A) \, d\Pi(x^*, z^*)$$

$$\leq \int \int 1_{C_{x^*, z^*}}(A) \, d\mathcal{A}(A) \, d\Pi(x^*, z^*).$$

Recalling the definition of the set $E$ above, we get

$$\int \int 1_{C_{x^*, z^*}}(A) \, d\mathcal{A}(A) \, d\Pi(x^*, z^*) \leq \int_E \mathbb{P}_{A \sim \mathcal{A}}\{\|A(x^* - z^*)\| > t\varepsilon\} \, d\Pi(x^*, z^*).$$

Using the definition of $C_0$, $E$ and $D$, we deduce that

$$I_1 \leq C_{\mathsf{abs}}(\varepsilon, t\varepsilon; \mathcal{A}, D). \quad \text{(C.12)}$$

Now we bound $I_2$. We further split the integral $I_2$ as follows:

$$I_2 = I_{2_1} + I_{2_2},$$   (C.13)

where

$$I_{2_1} = \int \int 1_{C^c_{x^*,z^*}}(A) \int 1_{B^c_{c\sigma}}(e) \int 1_{B_{2,\hat{x}}}(z^*) \, d\mathcal{P}(\cdot|Ax^* + e, A)(\hat{x}) \, d\mathcal{E}(e) \, d\mathcal{A}(A) \, d\Pi(x^*, z^*)$$

$$I_{2_2} = \int \int 1_{C^c_{x^*,z^*}}(A) \int 1_{B_{c\sigma}}(e) \int 1_{B_{2,\hat{x}}}(z^*) \, d\mathcal{P}(\cdot|Ax^* + e, A)(\hat{x}) \, d\mathcal{E}(e) \, d\mathcal{A}(A) \, d\Pi(x^*, z^*)$$

Let us first find an upper bound for $I_{2_1}$. We have

$$I_{2_1} = \int \int 1_{C^c_{x^*,z^*}}(A) \int 1_{B^c_{c\sigma}}(e) \int 1_{B_{2,\hat{x}}}(z^*) \, d\mathcal{P}(\cdot|Ax^* + e, A)(\hat{x}) \, d\mathcal{E}(e) \, d\mathcal{A}(A) \, d\Pi(x^*, z^*)$$

$$\leq \int \int 1_{C^c_{x^*,z^*}}(A) \int 1_{B^c_{c\sigma}}(e) \, d\mathcal{E}(e) \, d\mathcal{A}(A) \, d\Pi(x^*, z^*)$$

$$\leq \int 1_{B^c_{c\sigma}}(e) \, d\mathcal{E}(e),$$

and therefore, by Definition 2.4,

$$I_{2_1} = \mathcal{E}(B^c_{c\sigma}) \leq D_{\mathsf{upp}}(c\sigma; \mathcal{E}).$$   (C.14)

We now find a bound for $I_{2_2}$. We first use Definition 2.5 to write

$$I_{2_2} = \int \int 1_{C^c_{x^*,z^*}}(A) \int 1_{B_{c\sigma}}(e) p_{\mathcal{E}}(e) \int 1_{B_{2,\hat{x}}}(z^*) d\mathcal{P}(\cdot|Ax^* + e, A)(\hat{x}) \, de \, d\mathcal{A}(A) \, d\Pi(x^*, z^*).$$

Now define the new variable $e' = e + A(x^* - z^*)$. Since, in the integrand, $\|e\| \leq c\sigma$ (due to the indicator function $1_{B_{c\sigma}}(e)$) and $\|A(x^* - z^*)\| \leq t\varepsilon$ (due to the indicator function $1_{C^c_{x^*,z^*}}(A)$), Definition 2.5 yields the bound

$$I_{2_2} \leq D_{\mathsf{shift}}(t\varepsilon, c\sigma; \mathcal{E}) \int \int 1_{C^c_{x^*,z^*}}(A) \int 1_{B_{2\sigma}}(e' - A(x^* - z^*)) p_{\mathcal{E}}(e')$$

$$\times \int 1_{B_{2,\hat{x}}}(z^*) \, d\mathcal{P}(\cdot|Az^* + e', A)(\hat{x}) \, de' \, d\mathcal{A}(A) \, d\Pi(x^*, z^*).$$

We now drop the first two indicator functions and relabel the variables $e'$ and $\hat{x}$ as $e$ and $\hat{z}$, respectively, to obtain

$$I_{2_2} \leq D_{\mathsf{shift}}(t\varepsilon, c\sigma; \mathcal{E}) \int \int \int \int 1_{B_{2,\hat{z}}}(z^*) \, d\mathcal{P}(\cdot|Az^* + e, A)(\hat{z}) \, d\mathcal{E}(e) \, d\mathcal{A}(A) \, d\Pi(x^*, z^*).$$

This gives

$$I_{2_2} \leq D_{\mathsf{shift}}(t\varepsilon, c\sigma; \mathcal{E}) q,$$

where $q$ is as in (C.10). Combining this with (C.13), we deduce that

$$I_2 \leq D_{\mathsf{upp}}(c\sigma, \varepsilon) + D_{\mathsf{shift}}(t\varepsilon, c\sigma; \mathcal{E}) q.$$

To complete the proof, we combine this with (C.11) and (C.12), and then recall (C.10) once more.  □

### C.2.4  Decomposing distributions

The following lemma is in large part similar to [49, Lem. A.1]. However, we streamline and rewrite its proof for clarity and completeness, fix a number of small issues and make an addition to the statement (see item (v) below) that is important for proving our main result.

**Lemma C.7** (Decomposing distributions). *Let $\mathcal{R}, \mathcal{P}$ be arbitrary distributions on $\mathbb{R}^n$, $p \geq 1$ and $\eta, \rho, \delta > 0$. If $W_p(\mathcal{R}, \mathcal{P}) \leq \rho$ and $k \in \mathbb{N}$ is such that*

$$\min\{\log \mathrm{Cov}_{\eta,\delta}(\mathcal{P}), \log \mathrm{Cov}_{\eta,\delta}(\mathcal{R})\} \leq k,$$   (C.15)

*then there exist distributions $\mathcal{R}', \mathcal{R}'', \mathcal{P}', \mathcal{P}''$, a constant $0 < \delta' \leq \delta$ and a discrete distribution $\mathcal{Q}$ with $\mathrm{supp}(\mathcal{Q}) = S$ satisfying*

(i) $\min\{W_\infty(\mathcal{P}', \mathcal{Q}), W_\infty(\mathcal{R}', \mathcal{Q})\} \le \eta$,

(ii) $W_\infty(\mathcal{R}', \mathcal{P}') \le \frac{\rho}{\delta^{1/p}}$,

(iii) $\mathcal{P} = (1 - 2\delta')\mathcal{P}' + (2\delta')\mathcal{P}''$ and $\mathcal{R} = (1 - 2\delta')\mathcal{R}' + (2\delta')\mathcal{R}''$,

(iv) $|S| \le e^k$,

(v) and $S \subseteq \operatorname{supp}(\mathcal{P})$ if $\mathcal{P}$ attains the minimum in (C.15) with $S \subseteq \operatorname{supp}(\mathcal{R})$ otherwise.

This lemma states that two distributions that are close in Wasserstein $p$-distance, and for which at least one has small approximate covering number (C.15), can be decomposed into mixtures (iii) of distributions, where the following holds. One of the distributions, say $\mathcal{P}'$, is close (i) in Wasserstein-$\infty$ distance to a discrete distribution $\mathcal{Q}$ with the cardinality of its support (iv) bounded by the approximate covering number. The other $\mathcal{R}'$ is close in Wasserstein-$\infty$ distance to $\mathcal{P}'$. Moreover, both mixtures (iii) are dominated by these distributions: the 'remainder' terms $\mathcal{P}''$ and $\mathcal{R}''$ are associated with a small constant $\delta' \le \delta$, meaning they are sampled with probability $\le \delta$ when drawing from either $\mathcal{P}$ or $\mathcal{R}$. Note that if $p < \infty$ then the Wasserstein-$\infty$ distance between $\mathcal{R}'$ and $\mathcal{P}'$ may get larger as $\delta$ shrinks, i.e., as the remainder gets smaller. However, this does not occur when $p = \infty$, as (ii) is independent of $\delta$ in this case.

*Proof.* Without loss of generality, we assume that $\log \operatorname{Cov}_{\eta,\delta}(\mathcal{P}) \le k$. Then $\operatorname{Cov}_{\eta,\delta}(\mathcal{P}) \le e^k$ and hence there is a set $S = \{u_i\}_{i=1}^l \subseteq \operatorname{supp}(\mathcal{P})$ with $l \le e^k$, where the $u_i$ are the centres of the balls used to cover at least $1 - \delta$ of the measure of $\mathcal{P}$. That is,

$$\mathcal{P}\left[\bigcup_{i=1}^l B(u_i, \eta)\right] = \mathbb{P}_{x \sim \mathcal{P}}\left[x \in \bigcup_{i=1}^l B(u_i, \eta)\right] =: 1 - c^* \ge 1 - \delta.$$

We now define $f : \mathbb{R}^n \to \mathbb{R}$ so that $f(x) = 0$ if $x$ lies outside these balls, and otherwise, $f(x)$ is the equal to the reciprocal of the number of balls in which $x$ is contained. Namely,

$$f(x) = \begin{cases} \frac{1}{\sum_{i=1}^l \mathbf{1}_{B(u_i, \eta)}(x)} & \text{if } x \in \bigcup_{i=1}^l B(u_i, \eta) \\ 0 & \text{otherwise} \end{cases}.$$

We divide the remainder of the proof into a series of steps.

*1. Construction of $\mathcal{Q}'$.* We will now define a finite measure $\mathcal{Q}'$. The point of $\mathcal{Q}'$ is to, concentrate the mass of the measure $\mathcal{P}$ into the centres of the balls $u_i$. If the sets $B(u_i, \eta)$ are disjoint, then this is straightforward. However, to ensure that $\mathcal{Q}'$ is indeed a probability measure, we need to normalize and account for any non-trivial intersections. This is done via the function $f$. Pick some arbitrary $\hat{u} \notin \{u_1, \ldots, u_l\}$ and define

$$\mathcal{Q}' = \sum_{i=1}^l \left(\int_{B(u_i, \eta)} f(x) \, d\mathcal{P}(x)\right) \delta_{u_i} + c^* \delta_{\hat{u}}.$$

Observe that

$$\int d\mathcal{Q}'(x) = \sum_{i=1}^l \int_{B(u_i, \eta)} f(x) \, d\mathcal{P}(x) + c^* = \mathcal{P}\left(\bigcup_{i=1}^l B(u_i, \eta)\right) + c^* = (1 - c^*) + c^* = 1,$$

and therefore $\mathcal{Q}'$ is a probability distribution supported on the finite set $S \cup \{\hat{u}\}$.

*2. Coupling $\mathcal{Q}', \mathcal{P}$.* Now that we have associated all the mass of $\mathcal{P}$ with the points $u_i$, we can define a coupling $\Pi$ between $\mathcal{Q}'$ and $\mathcal{P}$ that associates the mass of $\mathcal{P}$ and $u_i$ with a single measure. Moreover, this measure will keep points within $\eta$ distance of each other with high probability. We define $\Pi$ as follows for measurable sets $E, F \subseteq \mathbb{R}^n$:

$$\Pi(E, F) = \sum_{i=1}^l \mathbf{1}_F(u_i) \int_{B(u_i, \eta) \cap E} f(x) \, d\mathcal{P}(x) + \mathbf{1}_F(\hat{u}) \mathcal{P}\left(E \setminus \bigcup_{i=1}^l B(u_i, \eta)\right).$$

To see that this is a coupling, we first observe that

$$\Pi(\mathbb{R}^n, F) = \sum_{i=1}^l \mathbf{1}_F(u_i) \int_{B(u_i, \eta)} f(x) \, d\mathcal{P}(x) + \mathbf{1}_F(\hat{u})(1 - c^*) \equiv \mathcal{Q}'(F),$$

which gives the result for the first marginal. For the other, we have

$$\Pi(E, \mathbb{R}^n) = \sum_{i=1}^{l} \int_{B(u_i, \eta) \cap E} f(x) \, d\mathcal{P}(x) + \mathcal{P}\left(E \backslash \bigcup_{i=1}^{l} B(u_i, \eta)\right).$$

By definition of $f$, this is precisely

$$\Pi(E, \mathbb{R}^n) = \mathcal{P}\left(E \cap \bigcup_{i=1}^{l} B(u_i, \eta)\right) + \mathcal{P}\left(E \backslash \bigcup_{i=1}^{l} B(u_i, \eta)\right) \equiv \mathcal{P}(E),$$

which gives the result for the second marginal. Note that $\Pi$ was only defined for product sets, but, since $\mathcal{Q}'$ is finitely supported, it follows directly from Lemma C.3 that it extends to arbitrary measurable sets in the product sigma-algebra. We now show that $\Pi[\|x_1 - x_2\| > \eta] \leq c^* \leq \delta$. That is, we show that most points drawn from $\Pi$ are within $\eta$ distance of each other. By law of total probability we have

$$\Pi(\|x_1 - x_2\| > \eta) = \sum_{i=1}^{l} \Pi(\|x_1 - x_2\| > \eta | x_2 = u_i) \Pi(x_2 = u_i)$$
$$+ \Pi(\|x_1 - x_2\| > \eta | x_2 = \hat{u}) \Pi(x_2 = \hat{u})$$
$$= \sum_{i=1}^{l} \Pi(U_i, u_i) \mathcal{Q}'(u_i) + \Pi(\hat{U}, \hat{u}) \mathcal{Q}'(\hat{u}),$$

where $U_i = \{x : \|x - u_i\| > \eta\}$ and $\hat{U} = \{x : \|x - \hat{u}\| > \eta\}$. Notice that $U_i \cap B(u_i, \eta) = \emptyset$ and therefore

$$\Pi(U_i, u_i) = \int_{\{x : \|x - u_i\| > \eta\} \cap B(u_i, \eta)} f \, d\mathcal{P} = 0.$$

Hence

$$\sum_{i=1}^{l} \Pi(U_i, u_i) \mathcal{Q}'(u_i) + \Pi(\hat{U}, \hat{u}) \mathcal{Q}'(\hat{u}) = \Pi(\hat{U}, \hat{u}) \mathcal{Q}'(\hat{u})$$

and, since $\mathcal{Q}'(\hat{u}) = c^*$, we have $\Pi(\hat{U}, \hat{u}) \mathcal{Q}'(\hat{u}) \leq c^* \leq \delta$. This gives

$$\Pi(\|x_1 - x_2\| > \eta) \leq \delta, \tag{C.16}$$

as required.

*3. Coupling $\mathcal{P}, \mathcal{R}$.* The next step is to introduce $\mathcal{R}$. With the assumption that $W_p(\mathcal{R}, \mathcal{P}) \leq \rho$, by definition there exists a coupling $\Gamma$ between $\mathcal{P}$ and $\mathcal{R}$ such that $\mathbb{E}_\Gamma[\|x_1 - x_2\|^p] \leq \rho^p$. Markov's inequality then gives that

$$\Gamma\left(\|x_1 - x_2\| \geq \frac{\rho}{\delta^{1/p}}\right) \leq \frac{\mathbb{E}_\Gamma[\|x_1 - x_2\|^p]}{\frac{\rho^p}{\delta}} \leq \delta. \tag{C.17}$$

*4. Coupling $\mathcal{P}, \mathcal{Q}', \mathcal{R}$.* We next couple $\mathcal{P}, \mathcal{Q}'$ and $\mathcal{R}$. Before doing so, we first discuss the goal of our final coupling. Recall that we have the distribution $\mathcal{P}$, the distribution $\Pi$ that couples $\mathcal{P}, \mathcal{Q}'$ closely except for up to $\delta$ mass of $\mathcal{P}$, and $\Gamma$ which keeps $\mathcal{P}, \mathcal{R}$ close again except for up to $\delta$ of the mass of $\Gamma$. We want to decompose $\mathcal{P}$ into the portions that are $\eta$ close to $\mathcal{Q}'$, and points that are not. These will become $\mathcal{P}'$ and $\mathcal{P}''$, respectively. At the same time, we want to decompose $\mathcal{R}$ to points that are $\frac{\rho}{\delta^{1/p}}$ close to $\mathcal{P}'$, and points that are not. Naturally this will become $\mathcal{R}'$ and $\mathcal{R}''$. To achieve this, we couple $\mathcal{P}, \mathcal{Q}'$ and $\mathcal{R}$ in this step and then use this to construct the final decomposition in the next step.

We have measures $\mathcal{P}, \mathcal{Q}'$ and $\mathcal{R}$ and couplings $\Pi$ of $\mathcal{P}, \mathcal{Q}'$ and $\Gamma$ of $\mathcal{P}, \mathcal{R}$. We will in a sense, couple $\Pi, \Gamma$. Since $(\mathbb{R}^n)^3$ is a Polish space, by [10, Lem. 8.4], there exists a coupling $\Omega$ with

$$\pi_{1,2} \sharp \Omega = \Pi, \quad \pi_{1,3} \sharp \Omega = \Gamma,$$

where $\pi_{1,2}(x_1, x_2, x_3) = (x_1, x_2)$ and likewise for $\pi_{1,3}$. One should intuitively think of the $x_1$ component as samples from $\mathcal{P}$, the $x_2$ component as samples from $\mathcal{Q}$, and the $x_3$ component as samples from $\mathcal{R}$. With the base measure defined, we still want to ensure that $x_1, x_3$ are sampled

closely, and $x_1, x_2$ are as well. Consider the event such that $x_1, x_3$ are $\frac{\rho}{\delta^{1/p}}$ close and $x_1, x_2$ are $\eta$ close: namely,

$$E := \{(x_1, x_2, x_3) : \|x_1 - x_3\| \le \rho/\delta^{1/p} \text{ and } \|x_1 - x_2\| \le \eta\}.$$

Split up the negation of the two events of $\|x_1 - x_3\| \le \rho/\delta^{1/p}$ and $\|x_1 - x_2\| \le \eta$ into the events

$$E_1 := \{(x_1, x_2, x_3) : \|x_1 - x_2\| > \eta, z \in \mathbb{R}^n\},$$
$$E_2 := \{(x_1, x_2, x_3) : \|x_1 - x_3\| > \rho/\delta^{1/p}, y \in \mathbb{R}^n\},$$

so that $E^c = E_1 \cup E_2$. We will now show $\Omega(E_1) \le \delta$. Write $E_1 = E_1' \times \mathbb{R}^n$ where $E_1' = \{(x_1, x_2) : \|x_1 - x_2\| > \eta\}$ satisfies $\Pi(E_1') \le \delta'$ by (C.16). Then

$$\delta' \ge \Pi(E_1') = \int 1_{E_1'}(x_1, x_2)\, \mathrm{d}\pi_{1,2}\sharp\Omega(x_1, x_2)$$

$$= \int 1_{E_1'}(p^{1,2}(x_1, x_2, x_3))\, \mathrm{d}\Omega(x_1, x_2, x_3)$$

$$= \int 1_{E_1}(x_1, x_2, x_3)\, \mathrm{d}\Omega(x_1, x_2, x_3)$$

$$= \Omega(E_1),$$

as required. Using (C.17), we also have the analogous result for $E_2$. Hence $\Omega(E^c) = \Omega(E_1 \cup E_2) \le \Omega(E_1) + \Omega(E_2) =: 2\delta' \le 2\delta'$, and consequently,

$$\Omega(E) = 1 - 2\delta' \ge 1 - 2\delta,$$
$$\text{where } E := \{(x_1, x_2, x_3) : \|x_1 - x_3\| \le \rho/\delta^{1/p} \text{ and } \|x_1 - x_2\| \le \eta\}. \tag{C.18}$$

*4. Decomposing $\mathcal{P}, \mathcal{R}$.* Finally, we define $\mathcal{P}', \mathcal{P}'', \mathcal{R}', \mathcal{R}''$ and $\mathcal{Q}$ by conditioning on the events $E$ and $E^c$, as follows:

$$\mathcal{P}'(A) = \Omega(A, \mathbb{R}^n, \mathbb{R}^n | E),$$
$$\mathcal{R}'(A) = \Omega(\mathbb{R}^n, \mathbb{R}^n, A | E),$$
$$\mathcal{P}''(A) = \Omega(A, \mathbb{R}^n, \mathbb{R}^n | E^c),$$
$$\mathcal{R}''(A) = \Omega(\mathbb{R}^n, \mathbb{R}^n, A | E^c),$$
$$\mathcal{Q}(A) = \Omega(\mathbb{R}^n, A, \mathbb{R}^n | E).$$

This gives

$$\mathcal{P}(A) = \Omega(A, \mathbb{R}^n, \mathbb{R}^n) = \Omega(E)\Omega(A, \mathbb{R}^n, \mathbb{R}^n | E) + \Omega(E^c)\Omega(A, \mathbb{R}^n, \mathbb{R}^n | E^c)$$
$$= (1 - 2\delta')\mathcal{P}'(A) + 2\delta'\mathcal{P}''(A)$$

and similarly

$$\mathcal{R}(A) = \Omega(\mathbb{R}^n, A, \mathbb{R}^n) = \Omega(E)\Omega(\mathbb{R}^n, A, \mathbb{R}^n | E) + \Omega(E^c)\Omega(\mathbb{R}^n, A, \mathbb{R}^n | E^c)$$
$$= (1 - 2\delta')\mathcal{R}'(A) + 2\delta'\mathcal{R}''(A).$$

We now claim that these distributions satisfy (i)-(v) in the statement of the lemma. We have already shown that (iii) holds. To show (i), we define a coupling $\gamma$ of $\mathcal{P}', \mathcal{Q}$ by $\gamma(B) = \Omega(B, \mathbb{R}^n | E)$ for any $B \subseteq (\mathbb{R}^n)^2$. Observe that $\gamma(A, \mathbb{R}^n) = \Omega(A, \mathbb{R}^n, \mathbb{R}^n | E) = \mathcal{P}'(A)$ and $\gamma(\mathbb{R}^n, A) = \Omega(\mathbb{R}^n, A, \mathbb{R}^n | E) = \mathcal{Q}(A)$. Hence this is indeed a coupling of $\mathcal{P}', \mathcal{Q}$. Therefore it suffices to show that $\gamma(B) = 0$, where $B$ is the event $\{\|x_1 - x_2\| > \eta\}$. We have $\gamma(B) = \Omega(B, \mathbb{R}^n | E)$. Recall that for $(x_1, x_2, x_3) \in E$, we have $\|x_1 - x_2\| \le \eta$. Hence $\Omega(B, \mathbb{R}^n | E) = 0$. Therefore, $W_\infty(\mathcal{P}', \mathcal{Q}) \le \eta$, which gives (i).

Similarly, for (ii) we define a coupling $\gamma'$ of $\mathcal{R}', \mathcal{P}'$ by $\gamma'(B) = \Omega(\overline{B} | E)$ where $\overline{B} := \{(x_1, x_2, x_3) : (x_1 x_3) \in B, x_2 \in \mathbb{R}^n\}$. With similar reasoning as the previous case, $\gamma'$ is a coupling of $\mathcal{R}', \mathcal{P}'$ and, for $(x_1, x_2, x_3) \in E$ we have $\|x_1 - x_3\| \le \rho/\delta^{1/p}$, so letting $B$ be the event $\{\|x_1 - x_3\| > \rho/\delta^{1/p}\}$, we conclude that $W_\infty(\mathcal{R}', \mathcal{P}') \le \rho/\delta^{1/p}$. This gives (ii).

Finally we verify (iv) and (v). First recall that both properties hold for $\mathcal{Q}'$ by construction. The results now follow from the fact that $\mathcal{Q}'(\cdot) = \Omega(\mathbb{R}^n, \cdot, \mathbb{R}^n)$ and $\mathcal{Q}(\cdot) = \Omega(\mathbb{R}^n, \cdot, \mathbb{R}^n | E)$. $\qquad \square$

## C.3  Proof of Theorem 3.1

We now prove Theorem 3.1. This follows a similar approach to that of [49, Thm. 3.4], but with a series of significant modifications to account for the substantially more general setup considered in this work. We also streamline the proof and clarify a number of key steps.

*Proof of Theorem 3.1.* By Lemma C.7, we can decompose $\mathcal{P}, \mathcal{R}$ into measures $\mathcal{P}', \mathcal{P}''$ and $\mathcal{R}', \mathcal{R}''$, and construct a finite distribution $\mathcal{Q}$ supported on a finite set $S$ such that

(i) $\min\{W_\infty(\mathcal{P}', \mathcal{Q}), W_\infty(\mathcal{R}', \mathcal{Q})\} \le \eta$,

(ii) $W_\infty(\mathcal{R}', \mathcal{P}') \le \varepsilon' := \frac{\varepsilon}{\delta^{1/p}}$,

(iii) $\mathcal{P} = (1 - 2\delta')\mathcal{P}' + (2\delta')\mathcal{P}''$ and $\mathcal{R} = (1 - 2\delta')\mathcal{R}' + (2\delta')\mathcal{R}''$ for some $0 \le \delta' \le \delta$,

(iv) $|S| \le e^k$,

(v) and $S \subseteq \mathrm{supp}(\mathcal{P})$ if $\mathcal{P}$ attains the minimum in (C.15) with $S \subseteq \mathrm{supp}(\mathcal{R})$ otherwise.

It is helpful to briefly recall the construction of these sets. Beginning with $\delta, \eta$ as parameters for the approximate covering numbers, the distribution $\mathcal{Q}$ concentrates $1 - \delta$ of the mass of $\mathcal{P}$ into the centres of the $\eta$-radius balls used. Then the distributions $\mathcal{P}', \mathcal{R}'$ are the measures $\mathcal{P}, \mathcal{R}$ within the balls. We now write

$$
\begin{aligned}
p := {}& \mathbb{P}_{x^* \sim \mathcal{R}, A \sim \mathcal{A}, e \sim \mathcal{E}, \hat{x} \sim \mathcal{P}(\cdot|y, A)}[\|x^* - \hat{x}\| \ge (c+2)\eta + (c+2)\sigma] \\
\le {}& \mathbb{P}_{x^* \sim \mathcal{R}, A \sim \mathcal{A}, e \sim \mathcal{E}, \hat{x} \sim \mathcal{P}(\cdot|y, A)}[\|x^* - \hat{x}\| \ge (c+2)\eta + (c+1)\sigma + \varepsilon'] \\
\le {}& 2\delta' + (1 - 2\delta')\mathbb{P}_{x^* \sim \mathcal{R}', A \sim \mathcal{A}, e \sim \mathcal{E}, \hat{x} \sim \mathcal{P}(\cdot|y, A)}[\|x^* - \hat{x}\| \ge (c+1)(\eta + \sigma) + \varepsilon'] \\
=: {}& 2\delta' + (1 - 2\delta')q.
\end{aligned}
\tag{C.19}
$$

Here, in the first inequality we used the fact that $\sigma \ge \varepsilon'$, and in the second, we used the decomposition $\mathcal{R} = (1 - 2\delta')\mathcal{R}' + 2\delta'\mathcal{R}''$ and the fact that $\delta' \le \delta$. We now bound $q$ by using Lemma C.6 to replace the distribution $\mathcal{R}'$ by the distribution $\mathcal{P}'$. Writing $u = Az^* + e$, this lemma and (ii) give that

$$
\begin{aligned}
q \le {}& C_{\mathsf{abs}}(\varepsilon', t\varepsilon'; \mathcal{A}, D) + D_{\mathsf{upp}}(c'\sigma; \mathcal{E}) + D_{\mathsf{shift}}(t\varepsilon', c'\sigma; \mathcal{E})r, \\
& \text{where } r = \mathbb{P}_{z^* \sim \mathcal{P}', A \sim \mathcal{A}, e \sim \mathcal{E}, \hat{z} \sim \mathcal{P}(\cdot|u, A)}[\|z^* - \hat{z}\| \ge (c+1)(\eta + \sigma)]
\end{aligned}
\tag{C.20}
$$

and $D = \{x^* - z^* : (x^*, z^*) \in \mathrm{supp}(\Pi)\}$, for $\Pi$ being the $W_\infty$-optimal coupling of $\mathcal{R}', \mathcal{P}'$ guaranteed by (ii). Lemma C.1 implies that $\mathrm{supp}(\Pi) \subseteq \mathrm{supp}(\mathcal{R}') \times \mathrm{supp}(\mathcal{P}')$ and therefore

$$
D \subseteq \mathrm{supp}(\mathcal{R}') - \mathrm{supp}(\mathcal{P}').
$$

Now (iii) implies that $\mathrm{supp}(\mathcal{P}') \subseteq \mathrm{supp}(\mathcal{P})$. Similarly, (iii) implies that $\mathrm{supp}(\mathcal{R}') \subseteq \mathrm{supp}(\mathcal{R})$. But Lemma C.2 and (ii) imply that $\mathrm{supp}(\mathcal{R}') \subseteq B_{\varepsilon'}(\mathrm{supp}(\mathcal{P}'))$. Therefore

$$
D \subseteq B_{\varepsilon'}(\mathrm{supp}(\mathcal{P})) \cap \mathrm{supp}(\mathcal{R}) - \mathrm{supp}(\mathcal{P}) = D_1,
$$

where $D_1$ as in (3.2).

We now bound $r$. Observe first that

$$
W_\infty(\mathcal{P}', \mathcal{Q}) \le \eta' := \eta + \varepsilon'.
$$

Indeed, from (i) either $W_\infty(\mathcal{P}', \mathcal{Q}) \le \eta$ or $W_\infty(\mathcal{R}', \mathcal{Q}) \le \eta$. In the former case, the inequality trivially holds. In the latter case, we can use the triangle inequality and (ii) to obtain the desired bound. This implies that there is a coupling $\Gamma$ of $\mathcal{P}', \mathcal{Q}$ with $\mathrm{esssup}_\Gamma \|x - y\| \le \eta'$. Fix $\tilde{z} \in S$ and, for any Borel set $E \subseteq \mathbb{R}^n$, define

$$
\Gamma_{\tilde{z}}(E) = \frac{\Gamma(E, \tilde{z})}{\mathcal{Q}(\tilde{z})}.
$$

Then it is readily checked that $\Gamma_{\tilde{z}}(\cdot)$ defines a probability measure. Note also that $\Gamma_{\tilde{z}}$ is supported on a ball of radius $\eta'$ around $\tilde{z}$, since $\mathrm{esssup}_\Gamma(\|x - y\|) \le \eta'$. Recall that $\Gamma$ is a coupling between $\mathcal{P}'$ and $\mathcal{Q}$. Let $E \subseteq \mathbb{R}^n$ be a Borel set. Then Lemma C.3 gives that

$$
\mathcal{P}'(E) = \Gamma(E, \mathbb{R}^n) = \sum_{\tilde{z} \in S} \Gamma((E, \mathbb{R}^n)_{\tilde{z}}, \tilde{z}) = \sum_{\tilde{z} \in S} \Gamma(E, \tilde{z}) = \sum_{\tilde{z} \in S} \Gamma_{\tilde{z}}(E)\mathcal{Q}(\tilde{z}).
$$

Therefore, we can express $\mathcal{P}'$ as the mixture

$$\mathcal{P}'(\cdot) = \sum_{\tilde{z} \in S} \Gamma_{\tilde{z}}(\cdot) \mathcal{Q}(\tilde{z}).$$

Define the event $E = \{\|z^* - \hat{z}\| \geq (c+1)(\eta + \sigma)\} \subseteq \mathbb{R}^n \times \mathbb{R}^n$ so that the probability $r$ defined in (C.20) can be expressed as

$$r = \mathbb{E}_{z^* \sim \mathcal{P}', A \sim \mathcal{A}, e \sim \mathcal{E}, \hat{z} \sim \mathcal{P}(\cdot|A,u)}[1_E].$$

Using the above expression for $\mathcal{P}'$ we now write

$$r = \int \int \int \int 1_E(z^*, \hat{z}) \, d\mathcal{P}(\cdot|A, u)(\hat{z}) \, d\mathcal{E}(e) \, d\mathcal{A}(A) \, d\mathcal{P}'(z^*)$$

$$= \int \int \int \int 1_E(z^*, \hat{z}) \, d\mathcal{P}(\cdot|A, u)(\hat{z}) \, d\mathcal{E}(e) \, d\mathcal{A}(A) \, d\left(\sum_{\tilde{z} \in S} \mathcal{Q}(\tilde{z}) \Gamma_{\tilde{z}}(\cdot)\right)(z^*)$$

$$= \sum_{\tilde{z} \in S} \mathcal{Q}(\tilde{z}) \left(\int \int \int \int 1_E(z^*, \hat{z}) \, d\mathcal{P}(\cdot|A, u)(\hat{z}) \, d\mathcal{E}(e) \, d\mathcal{A}(A) \, d\Gamma_{\tilde{z}}(z^*)\right),$$

where the last line holds as $\mathcal{Q}(\tilde{z})$ is a constant. Hence

$$r = \sum_{\tilde{z} \in S} \mathcal{Q}(\tilde{z}) \mathbb{P}_{z^* \sim \Gamma_{\tilde{z}}, A \sim \mathcal{A}, e \sim \mathcal{E}, \hat{z} \sim \mathcal{P}(\cdot|A,u)}[E]. \tag{C.21}$$

Now we bound each term in this sum. We do this by decomposing $\mathcal{P}$ into a mixture of three probability measures depending on $\tilde{z} \in S$. To do this, let $\theta = c(\eta + \sigma)$ and observe that, for any Borel set $E \subseteq \mathbb{R}^n$,

$$\mathcal{P}(E) = \mathcal{P}(E \cap B_\theta(\tilde{z})) + \mathcal{P}(E \cap B_\theta^c(\tilde{z}))$$

$$= \mathcal{P}(E \cap B_\theta(\tilde{z})) + \mathcal{P}(E \cap B_\theta^c(\tilde{z})) + (1 - 2\delta') \mathcal{Q}(\tilde{z}) \Gamma_{\tilde{z}}(E) - (1 - 2\delta') \mathcal{Q}(\tilde{z}) \Gamma_{\tilde{z}}(E)$$

$$= \mathcal{P}(E \cap B_\theta(\tilde{z})) - (1 - 2\delta') \mathcal{Q}(\tilde{z}) \Gamma_{\tilde{z}}(E \cap B_\theta(\tilde{z}))$$

$$\quad + \mathcal{P}(E \cap B_\theta^c(\tilde{z})) - (1 - 2\delta') \mathcal{Q}(\tilde{z}) \Gamma_{\tilde{z}}(E \cap B_\theta^c(\tilde{z}))$$

$$\quad + (1 - 2\delta') \mathcal{Q}(\tilde{z}) \Gamma_{\tilde{z}}(E).$$

Now define the constants

$$c_{\tilde{z},\mathrm{mid}} = \mathcal{P}(B_\theta(\tilde{z})) - (1 - 2\delta') \mathcal{Q}(\tilde{z}) \Gamma_{\tilde{z}}(B_\theta(\tilde{z})), \quad c_{\tilde{z},\mathrm{ext}} = \mathcal{P}(B_\theta^c(\tilde{z})) - (1 - 2\delta') \mathcal{Q}(\tilde{z}) \Gamma_{\tilde{z}}(B_\theta^c(\tilde{z})).$$

and let

$$\mathcal{P}_{\tilde{z},\mathrm{int}}(E) = \Gamma_{\tilde{z}}(E)$$

$$\mathcal{P}_{\tilde{z},\mathrm{mid}}(E) = \frac{1}{c_{\tilde{z},\mathrm{mid}}} \left(\mathcal{P}(E \cap B_\theta(z^*)) - (1 - 2\delta') \mathcal{Q}(z^*) \Gamma_{\tilde{z}}(E \cap B_\theta(z^*))\right),$$

$$\mathcal{P}_{\tilde{z},\mathrm{ext}}(E) = \frac{1}{c_{\tilde{z},\mathrm{ext}}} (\mathcal{P}(E \cap B_\theta^c(z^*)) - (1 - 2\delta') \mathcal{Q}(z^*) \Gamma_{\tilde{z}}(E \cap B_\theta^c(z^*))).$$

Then $\mathcal{P}$ can be expressed as the mixture

$$\mathcal{P} = (1 - 2\delta') \mathcal{Q}(\tilde{z}) \mathcal{P}_{\tilde{z},\mathrm{int}} + c_{\tilde{z},\mathrm{mid}} \mathcal{P}_{\tilde{z},\mathrm{mid}} + c_{\tilde{z},\mathrm{ext}} \mathcal{P}_{\tilde{z},\mathrm{ext}}. \tag{C.22}$$

To ensure this is a well-defined mixture, we need to show that $\mathcal{P}_{\tilde{z},\mathrm{mid}}$ and $\mathcal{P}_{\tilde{z},\mathrm{ext}}$ are probability measures. However, by (iii) we have, for any Borel set $E \subseteq \mathbb{R}^n$,

$$\mathcal{P}(E) \geq (1 - 2\delta') \mathcal{P}'(E) = (1 - 2\delta') \sum_{\tilde{z} \in S} \Gamma_{\tilde{z}}(E) \mathcal{Q}(\tilde{z}) \geq (1 - 2\delta') \Gamma_{\tilde{z}}(E) \mathcal{Q}(\tilde{z}).$$

Therefore, $\mathcal{P}_{\tilde{z},\mathrm{mid}}$ and $\mathcal{P}_{\tilde{z},\mathrm{ext}}$ are well-defined, provided the constants $c_{\tilde{z},\mathrm{int}}, c_{\tilde{z},\mathrm{ext}} > 0$. However, if one of these constants is zero, then we can simply exclude this term from the mixture (C.22). For the rest of the theorem, we will assume that, at least, $c_{\tilde{z},\mathrm{ext}} > 0$.

It is now useful to note that

$$\mathrm{supp}(\mathcal{P}_{\tilde{z},\mathrm{mid}}) \subseteq B_\theta(\tilde{z}) \quad \text{and} \quad \mathrm{supp}(\mathcal{P}_{\tilde{z},\mathrm{ext}}) \subseteq B_\theta^c(\tilde{z}),$$

which follows immediately from their definitions, and also that
$$\text{supp}(\mathcal{P}_{\tilde{z},\text{int}}) \subseteq B_{\eta'}(\tilde{z}) \subseteq B_\theta(\tilde{z}).$$
where in the second inclusion we used the fact that $\eta' = \eta + \varepsilon/\delta^{1/p} \leq \eta + \sigma \leq c(\eta + \sigma) = \theta$, as $\sigma \geq \varepsilon/\delta^{1/p}$ and $c \geq 1$.

We now return to the sum (C.21). Consider an arbitrary term. First, observe that, for $z^* \sim \mathcal{P}$, we have $\mathbb{P}(z^* \sim \Gamma_{\tilde{z}}) = \mathcal{Q}(\tilde{z})(1 - 2\delta')$ by (C.22). Hence

$$\mathcal{Q}(\tilde{z})\mathbb{E}_{z^* \sim \Gamma_{\tilde{z}}, \hat{z} \sim \mathcal{P}(\cdot|A,u)}[1_E] = \frac{\mathbb{P}(z^* \sim \Gamma_{\tilde{z}})}{1 - 2\delta'}\mathbb{E}_{z^* \sim \mathcal{P}, \hat{z} \sim \mathcal{P}(\cdot|A,u)}[1_E | z^* \sim \Gamma_{\tilde{z}}]$$

$$= \frac{\mathbb{P}(z^* \sim \Gamma_{\tilde{z}})}{(1 - 2\delta')}\frac{1}{\mathbb{P}(z^* \sim \Gamma_{\tilde{z}})}\int\int 1_E 1_{z^* \sim \Gamma_{\tilde{z}}}\,\mathrm{d}\mathcal{P}(z^*)\,\mathrm{d}\mathcal{P}(\cdot|A,u)(\hat{z}).$$

Recall that $z^* \sim \Gamma_{\tilde{z}}$ is supported in $B_{\eta'}(\tilde{z})$. Therefore, for the event $E$ to occur, i.e., $\|z^* - \hat{z}\| > (c+1)(\eta + \sigma)$, it must be that $\hat{z} \in B_\theta^c(\tilde{z})$, which means that $\hat{z} \sim \mathcal{P}_{\tilde{z},\text{ext}}(\cdot|A,u)$. Hence

$$\mathcal{Q}(\tilde{z})\mathbb{E}_{z^* \sim \Gamma_{\tilde{z}}, \hat{z} \sim \mathcal{P}(\cdot|A,u)}[1_E] \leq \frac{1}{1 - 2\delta'}\int\int 1_{\hat{z} \sim \mathcal{P}_{\tilde{z},\text{ext}}(\cdot|A,u)}1_{z^* \sim \Gamma_{\tilde{z}}}\,\mathrm{d}\mathcal{P}(z^*)\,\mathrm{d}\mathcal{P}(\cdot|A,u)(\hat{z})$$

$$= \frac{1}{1 - 2\delta'}\mathbb{P}[z^* \sim \mathcal{P}_{\tilde{z},\text{int}}, \hat{z} \sim \mathcal{P}_{\tilde{z},\text{ext}}(\cdot|A,u)].$$

Now fix $A \in \mathbb{R}^{m \times n}$. Let $\mathcal{H}_{\tilde{z},\text{int},A}$ be the distribution of $y^* = Az^* + e$ for $z^* \sim \mathcal{P}_{\tilde{z},\text{int}}$ and $e \sim \mathcal{E}$ independently, and define $\mathcal{H}_{\tilde{z},\text{ext},A}$ similarly. Then, by Fubini's theorem, we have

$$\mathcal{Q}(\tilde{z})\mathbb{E}_{z^* \sim \Gamma_{\tilde{z}}, A \sim \mathcal{A}, e \sim \mathcal{E}, \hat{z} \sim \mathcal{P}(\cdot|A,u)}[1_E] \leq \frac{1}{1 - 2\delta'}\mathbb{E}_{A \sim \mathcal{A}, e \sim \mathcal{E}}\mathbb{P}[y^* \sim \mathcal{H}_{\tilde{z},\text{int},A}, \hat{y} \sim \mathcal{H}_{\tilde{z},\text{ext},A}(\cdot|y^*)].$$

Now let $\mathcal{H}_{\tilde{z},A}$ be the distribution of $y = Az + e$ for $z \sim \mathcal{P}$ and $e \sim \mathcal{E}$ independently. Then Lemma C.4 (with $\mathcal{H} = \mathcal{H}_{\tilde{z},A}$, $\mathcal{H}_1 = \mathcal{H}_{\tilde{z},\text{int},A}$, $\mathcal{H}_2 = \mathcal{H}_{\tilde{z},\text{mid},A}$, $\mathcal{H}_3 = \mathcal{H}_{\tilde{z},\text{ext},A}$ and $a_1 = (1 - 2\delta')\mathcal{Q}(\tilde{z})$, $a_2 = c_{\tilde{z},\text{mid}}$, $a_3 = c_{\tilde{z},\text{ext}}$) gives

$$\mathcal{Q}(\tilde{z})\mathbb{E}_{z^* \sim \Gamma_{\tilde{z}}, A \sim \mathcal{A}, e \sim \mathcal{E}, \hat{z} \sim \mathcal{P}(\cdot|A,u)}[1_E] \leq \frac{1}{1 - 2\delta'}\mathbb{E}_{A \sim \mathcal{A}}\left[1 - \text{TV}(\mathcal{H}_{\tilde{z},\text{int},A}, \mathcal{H}_{\tilde{z},\text{ext},A})\right].$$

Finally, summing over all $\tilde{z}$ we deduce that

$$r = \sum_{\tilde{z} \in S}\mathcal{Q}(\tilde{z})\mathbb{E}_{z^* \sim \Gamma_{\tilde{z}}, A \sim \mathcal{A}, e \sim \mathcal{E}, \hat{z} \sim \mathcal{P}(\cdot|A,u)}[1_E] \leq \frac{1}{1 - 2\delta'}\sum_{\tilde{z} \in S}\mathbb{E}_{A \sim \mathcal{A}}[1 - \text{TV}(\mathcal{H}_{\tilde{z},\text{int},A}, \mathcal{H}_{\tilde{z},\text{ext},A})].$$
$$\text{(C.23)}$$

Now recall that $\mathcal{H}_{\tilde{z},\text{int},A}$ is the pushforward of a measure $\mathcal{P}_{\tilde{z},\text{int}}$ supported in $B_{\eta'}(\tilde{z})$, where $\eta' = \eta + \varepsilon' \leq \eta + \sigma$ and $\mathcal{H}_{\tilde{z},\text{ext},A}$ is the pushforward of a measure $\mathcal{P}_{\tilde{z},\text{ext}}$ supported in $B_\theta^c(\tilde{z})$, where $\theta = c(\eta + \sigma) \geq \frac{c}{2}(\eta' + \sigma)$. Therefore, Lemma C.5 (with $c$ replaced by $c/2$) gives that

$$\mathbb{E}_{A \sim \mathcal{A}}[1 - \text{TV}(\mathcal{H}_{\tilde{z},\text{int},A}, \mathcal{H}_{\tilde{z},\text{ext},A})] \leq C_{\text{low}}\left(\frac{2\sqrt{2}}{\sqrt{c}}; \mathcal{A}, D_{\tilde{z},\text{ext}}\right) + C_{\text{upp}}\left(\frac{\sqrt{c}}{2\sqrt{2}}; \mathcal{A}, D_{\tilde{z},\text{int}}\right)$$

$$+ 2D_{\text{upp}}\left(\frac{\sqrt{c}\sigma}{2\sqrt{2}}; \mathcal{E}\right),$$

where $D_{\tilde{z},\text{ext}} = \{x - \tilde{z} : x \in \text{supp}(\mathcal{P}_{\tilde{z},\text{ext}})\}$ and $D_{\tilde{z},\text{int}} = \{x - \tilde{z} : x \in \text{supp}(\mathcal{P}_{\tilde{z},\text{int}})\}$. It follows immediately from (C.22) that

$$\text{supp}(\mathcal{P}_{\tilde{z},\text{int}}), \text{supp}(\mathcal{P}_{\tilde{z},\text{ext}}) \subseteq \text{supp}(\mathcal{P}).$$

Moreover, $\tilde{z} \in S$ and therefore

$$D_{\tilde{z},\text{ext}}, D_{\tilde{z},\text{int}} \subseteq D_2,$$

where $D_2$ is as in (3.3). Using this, the previous bound and (C.23), we deduce that

$$r \leq \frac{|S|}{1 - 2\delta'}\left[C_{\text{low}}\left(\frac{2\sqrt{2}}{\sqrt{c}}; \mathcal{A}, D_2\right) + C_{\text{upp}}\left(\frac{\sqrt{c}}{2\sqrt{2}}; \mathcal{A}, D_2\right) + 2D_{\text{upp}}\left(\frac{\sqrt{c}\sigma}{2\sqrt{2}}; \mathcal{E}\right)\right].$$

To complete the proof, now substitute this into (C.19) and (C.20), to obtain
$$p \leq 2\delta' + [C_{\text{abs}}(\varepsilon', t\varepsilon'; \mathcal{A}, D_1) + D_{\text{upp}}(c'\sigma; \mathcal{E})]$$

$$+ 2D_{\text{shift}}(t\varepsilon', c'\sigma; \mathcal{E})|S|\left[C_{\text{low}}\left(\frac{2\sqrt{2}}{\sqrt{c}}; \mathcal{A}, D_2\right) + C_{\text{upp}}\left(\frac{\sqrt{c}}{2\sqrt{2}}; \mathcal{A}, D_2\right) + 2D_{\text{upp}}\left(\frac{\sqrt{c}\sigma}{2\sqrt{2}}; \mathcal{E}\right)\right].$$

The result now follows after recalling (iv), i.e., $|S| \leq e^k$, and the fact that $\delta' \leq \delta \leq 1/4$. $\qquad\square$

## C.4 Proof of Theorem 1.1

Finally, we now show how Theorem 3.1 implies the simplified result, Theorem 1.1.

*Proof of Theorem 1.1.* Let $p = \mathbb{P}\left[\|x^* - \hat{x}\| \geq (8d^2 + 2)(\eta + \sigma)\right]$. We use Theorem 3.1 with $\varepsilon$ replaced by $\varepsilon/(2m\theta)$. Let $c = 8d^2$, $c' = 2$, $t = \theta$ and $\varepsilon' = \varepsilon/(2\delta^{1/p}m\theta)$. Then Theorem 3.1 gives that

$$p \lesssim \delta + C_{\mathsf{abs}}(\varepsilon', \theta\varepsilon'; \mathcal{A}, D_1) + D_{\mathsf{upp}}(2\sigma; \mathcal{E})$$
$$+ 2D_{\mathsf{shift}}(\theta\varepsilon', 2\sigma; \mathcal{E})\mathrm{e}^k \left[C_{\mathsf{low}}\left(1/d; \mathcal{A}, D_2\right) + C_{\mathsf{upp}}\left(d; \mathcal{A}, D_2\right) + 2D_{\mathsf{upp}}(d\sigma; \mathcal{E})\right],$$

where

$$D_1 = B_{\varepsilon'}(\mathrm{supp}(\mathcal{P})) \cap \mathrm{supp}(\mathcal{R}) - \mathrm{supp}(\mathcal{P}) \subseteq B_{\varepsilon'}(\mathrm{supp}(\mathcal{P}) - \mathrm{supp}(\mathcal{P})),$$

$D_2 = D = \mathrm{supp}(\mathcal{P}) - \mathrm{supp}(\mathcal{P})$ and $k = \lceil \log \mathrm{Cov}_{\eta,\delta}(\mathcal{P}) \rceil$. Now since $\|Ax\| \leq \|A\|\|x\| \leq \theta\|x\|$, $\forall x \in \mathbb{R}^n$, we make take $C_{\mathsf{abs}}(\varepsilon', \theta\varepsilon'; \mathcal{A}, D_1) = 0$. Moreover, by Lemma B.1, we have

$$D_{\mathsf{shift}}(\theta\varepsilon', 2\sigma; \mathcal{E}) \leq \exp\left(\frac{m(4\sigma + \theta\varepsilon')}{2\sigma^2}\theta\varepsilon'\right) = \exp\left(\frac{\varepsilon}{\delta^{1/p}\sigma} + \frac{\varepsilon^2}{8\delta^{2/p}\sigma^2m}\right) \lesssim 1$$

where we used the facts that $m \geq 1$ and $\sigma \geq \varepsilon/\delta^{1/p}$. Hence

$$p \lesssim \delta + \mathrm{e}^k \left[C_-\left(1/d; \mathcal{A}, D_2\right) + C_+\left(d; \mathcal{A}, D_2\right) + C(d\sigma; \mathcal{E})\right].$$

Finally, Lemma B.1 implies that

$$D_{\mathsf{upp}}(d\sigma; \mathcal{E}) \leq \left(d\mathrm{e}^{1-d}\right)^{m/2} \leq \exp(-m/16),$$

where in the final step we used the fact that $d \geq 2$. This gives the result. $\qquad\square$

