# OpenReview forum: "How many measurements are enough? Bayesian recovery in inverse problems with general distributions"
_NeurIPS.cc/2025/Conference — NeurIPS 2025 spotlight_

### Official Review · Reviewer_h9UA · 2025-06-28

**Clarity:** 3
**Significance:** 3
**Originality:** 3
**Rating:** 5
**Confidence:** 3

**Summary:**

The authors provide a number of theoretical results with respect to sample complexity in Bayesian recovery. Namely, they show that sample complexity depends on the the intrinsic complexity of the prior and the concentration bounds for the forward operator and noise distributions. Overall, the authors unify prior theoretical work providing guarantees for the sample complexity of solving Bayesian inverse problems.

**Questions:**

See strengths and weaknesses.

**Ethical Concerns:**

["NO or VERY MINOR ethics concerns only"]

**Final Justification:**

After reading other reviews + all rebuttals, I will keep my score the same. This is an impactful theoretical work worthy of acceptance.

**Limitations:**

yes

**Paper Formatting Concerns:**

No concerns.

**Quality:**

3

**Strengths And Weaknesses:**

**Strengths**

S1) The paper is very well-written and easy to follow.

S2) The authors present a set of theoretical results which are thorough and (as far as I can tell) correct.

S3) The theoretical contribution is meaningful and worthy of publication at NeurIPS.

**Weaknesses**
W1) While I understand that this is work is entirely a theoretical contribution, it would still be worthwhile to show simple experiments that demonstrate the proposed theory in practice. While this is not necessarily required, it would be further elevate the work.

---

> ### Author Rebuttal · Authors · 2025-07-30
>
> We very much appreciate your effort in reviewing our paper and are delighted by your positive assessment. Here we address the comments in your report.
>
> ## W1) While I understand that this is work is entirely a theoretical contribution, it would still be worthwhile to show simple experiments that demonstrate the proposed theory in practice. While this is not necessarily required, it would be further elevate the work.
>
> Thanks for this comment. We wholeheartedly agree, and we are about to commence working on numerical implementations and developing a series of numerical experiments that demonstrate our theoretical results. We hope to report on these in the near future. However, we do not think doing so is feasible for this paper, given the tight deadlines and the space limitations. A particular issue is that practical posterior sampling algorithms are still very much an active area of inquiry. Please see our reply to reviewer b54M for more on this point. As a consequence, numerical implementations of posterior sampling are not 'simple', and therefore one has to be careful to ensure empirical verification of the theory is not obfuscated by numerical issues, e.g. lack of convergence of algorithms, parameter choices, and so forth. For this reason, we prefer to take more time to conduct a more rigorous and comprehensive study.
>
> Nonetheless, we agree this is an vital topic for future work. **We will add some more discussion on this in the final paragraph of §5 ("Limitations an future work").**

---

> > ### Comment · Reviewer_h9UA · 2025-08-02
> >
> > My thanks to the authors for addressing my question. I agree that, given the theoretical nature of the work, it is not necessary (or potentially feasible) to deliver empirical results during the rebuttal/discussion periods. Therefore, based on the other reviews and author replies, I will keep my score as it is.

---

> > > ### Author Response · Authors · 2025-08-04
> > > **Thanks!**
> > >
> > > Thanks for your reply and your review. We are delighted you will maintain your score as is.

---

### Official Review · Reviewer_maR6 · 2025-07-02

**Clarity:** 3
**Significance:** 4
**Originality:** 3
**Rating:** 5
**Confidence:** 4

**Summary:**

This paper studies the sample complexity to recover the true parameter with controlled accuracy in linear Bayesian inverse problems. It gives the sufficient conditions for Bayesian recovery for arbitrary real and approximate prior distributions, and forward and noise distributions. It establishes a non-asymptotic bound for sample complexity depending on the prior complexity (approximate covering number) and the concentration bounds for the forward and noise distributions. Then it applies to particular scenarios such as subgaussian measurements and subsampled orthogonal transforms with specific bounds.

**Questions:**

Here are a few minor questions/suggestions:

* line 280, why $\mathbb{E}(A^*A)=I$? What I can tell $A^*A$ has at most rank $q$. To show $A$ is isotropic, $\sqrt{n/m}$ ensures $\mathrm{tr}(\mathbb{E}(A^*A))=\mathbb{E}(q)=m$, which is enough. I do not think $\mathbb{E}(A^*A)=I$ holds or is needed.
* Bring the definition of 'near-optimality' from line 318 to line 106 when it was mentioned for the first time.
* line 216: based on lemmas B.2 and B.3, did you mean $\log(C_{low}), \log(C_{upp}) = \mathcal O(t^{-2})$? Is a symbol $\mathcal O$ missing in front of $(t-1)^{-2}$?
* $\varepsilon$ is not defined in Theorem 1.1. Add one before $\eta>0$ so it reads as $\varepsilon, \eta>0$.
* line 95: typo: 'achieve' $\to$ 'achieved'.
* line 163: 'be' and 'arise' are duplicate, remove one.
* line 209: $C_-$ $\to$ $C_{low}$.
* line 213: $\mathbb{P}$ $\to$ $\mathbb{P}_{A\sim \mathcal A}$.
* line 253: $C_{low}(2\sqrt{2}/\sqrt{2};\mathcal{A}, D_2)$ $\to$ $C_{low}(2\sqrt{2}/\sqrt{c};\mathcal{A}, D_2)$.
* line 584: $x\in B_\eta(x_i)$ $\to$ $z\in B_\eta(z_i)$.

**Ethical Concerns:**

["NO or VERY MINOR ethics concerns only"]

**Final Justification:**

Very good theory paper. I concur with other reviewers on the score and keep it.

**Limitations:**

yes.

**Quality:**

3

**Strengths And Weaknesses:**

# Strengths:

* This paper address the core question in Bayesian (linear) inverse problems on how many samples required to recover the truth.
* The sample complexity bounds apply to general real and approximate prior distributions, and forward and noise distributions.
* Its conclusion could potentially help resolve the "hallucinations" in deep learning.

# Weaknesses:
* Section 1.2 seems redundant given Sections 3.1 and 3.2.
* It should be properly emphasized that concentration bounds ($C\_{low}, C\_{upp}$) exponentially decay. The probability upper bound right above (3.2) appears to be large (notice the $e^k$ term) and readers might doubt whether it is a useful bound (smaller than 1).
* BTW, I understand the bounds are non-asymptotic; but it might help with the understanding if the authors could elaborate the asymptotic behavior of these probability bounds (e.g. $\delta$) when $m\to \infty$ or $m\to n$? Do they tend to 0?
* Difference set $\Sigma-\Sigma$ (line 252) is not a standard notation and should be explicitly defined. I assume the author(s) meant for $\\{x-y: x, y\in \Sigma\\}$, not $\Sigma\backslash\Sigma=\emptyset$ as interpreted by convention. There are instances like this spread over the paper without a clear definition.

---

> ### Author Rebuttal · Authors · 2025-07-30
>
> We very much appreciate your effort in reviewing our paper and are delighted by your positive assessment. Here we address the comments in your report.
>
> # Weaknesses:
>
> ## Section 1.2 seems redundant...
>
> Thanks for this comment. We agree there is overlap between the sections. The aim of §1.2 is to present simplified versions of the main results that the reader can more easily digest without reading all the technical setup. §3.1,3.2 present the main results in full technical detail. We feel this style of presentation, despite some redundancy, makes it easier for the reader. It is also quite common, in our experience, for papers to adopt this style. For this reason, we prefer to keep it.
>
> ## It should be properly emphasized...
>
> Excellent comment. This was also mentioned by reviewer b54M. **We will add the following remark about this immediately after Theorem 3.1:**
>
> "Since Theorem 3.1 considers arbitrary measurement $\mathcal{A}$ and noise $\mathcal{E}$ distributions, the number of measurements $m$ does not explicitly enter the bound. For typical distributions, a dependence on $m$ is found in the concentration bounds $C_{\mathsf{low}},C_{\mathsf{upp}},D_{\mathsf{upp}}$, as well as the terms $D_{\mathsf{shift}}$ and $C_{\mathsf{abs}}$. In particular, the former decay exponentially-fast in $m$ for the examples introduced in §1.2 and therefore compensate for the exponentially-large scaling in $k$ in the main bound. See §B for precise estimates. See also the discussion in §3.1 and 3.2 below."
>
> ## BTW, I understand the bounds are non-asymptotic...
>
> Good comment. **We will add the following after Theorem 3.5:**
>
> "It is worth at this stage discussing how the bound (3.4) behaves with respect to the various parameters. First, suppose that $\eta$ decreases so that the error bound becomes smaller. Then $\mathrm{Cov}_{\eta, \delta}(\cdot)$ increases, meaning, as expected, that more measurements are required to meet (3.4). Second, suppose that $\delta$ decreases, so that the failure probability shrinks. Then $\mathrm{Cov} (\cdot)$ and $\log(1 /\delta)$ both increase, meaning, once again, that more measurements are needed for (3.4) to hold. Both behaviours are as expected."
>
> You raise an interesting point regarding the regime $m \approx n$. Intuitively, one expects good recovery here, as there are as many, or even more, measurements $m$ as unknowns $n$. This is not reflected by our analysis. Our analysis is intended for the highly underdetermined regime, i.e., where the RHS of (3.4) is much smaller than $n$. However, it is conceivable that the RHS of (3.4) could be much larger than $n$ if, say, $\mathcal{R},\mathcal{P}$ had large approximate covering numbers.
> We believe it may be possible to derive a bound that covers both regimes $m \ll n$ (our bound) and $m \approx n$. This is an interesting question for future work.
>
>
>
>
> ## Difference set $\Sigma - \Sigma$ (line 252) is not...
>
> Thanks for pointing this out. We do mean $\Sigma - \Sigma = \{ x - z : x,z \in \Sigma \}$. We agree this could be misinterpreted. Since this notation is first used in line 50, **we will add the following there:**
>
> "Here, and throughout this paper, we write $S - S = \{ x_1 - x_2 : x_1,x_2 \in S \} \subseteq \mathbb{R}^n$ for the difference set associated with a set $S \subseteq \mathbb{R}^n$."
>
> ## There are instances like this...
>
> Thanks. We have checked the paper and found several other instances. In particular, the norm $\lVert  \cdot \rVert $ in the equation below line 49 is not defined until later (it is the Euclidean norm) and the norm $\lVert \cdot \rVert_{\infty}$ in the equation below line 93 is not defined (it is the infinity-norm). Also, we ought to state explicitly in line 50 that $\mathrm{supp}(\mathcal{P})$ is the support of the measure $\mathcal{P}$. **We will make these changes. We will also carefully check through the whole paper to make sure everything is properly defined.**
>
> # Questions:
>
> ## line 280, why $\mathbb{E}(A^* A) = I$?
>
> We believe the factor $\sqrt{n/m}$ is needed. With this factor, we can write
>
> $$
> \mathbb{E}(A^\top A) = \frac{n}{m} \mathbb{E} \left( \sum_{j=1}^{q} u_{i_j} u^\top_{i_j} \right) = \frac{n}{m} \mathbb{E} \left( \sum_{i=1}^{n} X_i u_i u^\top_i \right)
> $$
>
> where $X_i \sim \mathrm{Ber}(m/n)$, as in Definition 3.6. Since $\mathbb{E}(X_i) = m/n$, we get
>
> $$
> \mathbb{E}(A^\top A) = \frac{n}{m} \sum_{i=1}^{n} \frac{m}{n} u_i u^\top_i = \sum_{i=1}^{n} u_i u^\top_i = U^\top U = I
> $$
>
> as required. We may have misunderstood your question. **Please let us know if we can provide further clarification.**
>
>
>
> ## Bring the definition of 'near-optimality'...
>
> Thanks. **We will do this.**
>
> ## line 216: based on lemmas B.2....
>
> Apologies. We discovered this result is incorrect. For $C_{\mathsf{upp}}(t ; \mathcal{A} , \mathbb{R}^n)$ one does indeed have $C_{\mathsf{upp}}(t ; \mathcal{A} , \mathbb{R}^n) = O(t^{-2})$ for any (isotropic) distribution $\mathcal{A}$. This follows straightforwardly from Markov's inequality. However, one does not necessarily have $C_{\mathsf{low}}(1/t ; \mathcal{A}, \mathbb{R}^n) \rightarrow 0$ as $t \rightarrow \infty$ without further assumptions. **We will delete this sentence.**
>
>
> ## $\varepsilon$ is not defined in Theorem 1.1...
>
> Thanks. **We will add this.**
>
> ## Other comments
>
> Thanks for pointing out these typos. **We will fix these.**

---

### Official Review · Reviewer_b54M · 2025-07-02

**Clarity:** 3
**Significance:** 3
**Originality:** 3
**Rating:** 5
**Confidence:** 4

**Summary:**

This work seeks to quantify when, for linear inverse problems (i.e. compressive sensing), adequate Bayesian recovery is feasible in terms of the dimensionality reduction between the observation and original signal (i.e. the number of measurements). Assuming an approximate prior with low complexity and concentration bounds on the measurement and noise distribution, this paper upper bounds the probability that a sample from the posterior deviates too far from the original signal. Given this main theorem, the authors give results in specific instantiations, such as for sub-gaussian measurement designs and random sub-sampled orthogonal transforms.

**Questions:**

- When the prior is multimodal, with say $d$ modes, the covering number should also be $d$, assuming the modes can be contained in an $\eta$-sized radius. Is it correct to interpret Theorem 3.5 as saying roughly $\log{d}$ measurements are needed to specify what mode the signal was drawn from? Is there an intuitive way to see why this is the case?

- Line 328; is there any practical benefit to requiring trained DNNs have small coherence?

- It is worth reflecting on the literature around information-theoretically optimal recovery of $\hat{x}$ with algorithms like approximate message passing (AMP), which also map out the hardness of linear inverse problems through statistical-computational gaps (some references: "The dynamics of message passing on dense graphs, with applications to compressed sensing", "Message-passing algorithms for compressed sensing", and "Unrolled denoising networks provably learn to perform optimal Bayesian inference"). For example, in the asymptotic limit, how do your worst case error bounds on $\hat{x}$ relate to what is information theoretically possible for the minimal $||x^* - \hat{x}||$ (for a given $m$)?

- As a matter of clarity, it would be helpful to isolate (in a remark) where the bound presented in Theorem 3.1 depends on the measurement number: e.g., in $C_{\text\rm low}$ and $C_{\text \rm upp}$. It would also be helpful to include a brief sketch of the main ideas of the proof (especially where the different regions $\mathcal{D}$ come from, )

- Wherever you have $\text{supp}(\mathcal{P}) - \text{supp}(\mathcal{P})$ (lines 237, 291, etc.), do you mean $\text{supp}(\mathcal{P}) - \text{supp}(\mathcal{R})$?

**Ethical Concerns:**

["NO or VERY MINOR ethics concerns only"]

**Final Justification:**

- After reviewing the clarifications provided by the authors, I have a strong sense of the importance of this work as a fundamental theoretical contribution providing sample complexity bounds for Bayesian recovery and its broad applicability to posterior sampling algorithms. I have raised my score to a 5.

**Limitations:**

Limitations were clearly discussed.

**Quality:**

3

**Strengths And Weaknesses:**

Strengths:
- This paper gives quite universal theoretical results that quantify the measurements needed for the posterior to be concentrated around the original signal in linear inverse problems, abstracting approximate priors, measurement designs, and error distributions.
- The paper answers unsolved theoretical questions in the literature regarding sample complexity for recovery for subsampled orthogonal transforms.


Weaknesses:
- Since the paper gives sample complexity bounds for Bayesian recovery but doesn't approach, e.g., algorithms for actually sampling or estimating $\hat{x}$, the practical scope is limited.

---

> ### Author Rebuttal · Authors · 2025-07-30
>
> Thank you for your careful review our paper and your positive assessment of it. Here we address the comments in your report.
>
> # Weaknesses:
>
> ## Since the paper…
>
> Excellent point. Indeed, we do not discuss any examples of posterior sampling algorithms. There is a considerable amount of literature on these methods, which have been mainly studied empirically. Our paper aims to provide theory that these supports these empirical results. As an example, one popular class of methods involves Langevin dynamics. Since this requires computing $\nabla_x \log \mathcal{P}(x|y) = \nabla_x \log \mathcal{P}(y|x) + \nabla_x \log \mathcal{P}(x)$, a common approach is to use a score-based diffusion model which has access to an approximation of $\nabla_x \log \mathcal{P}(x)$. We refer to [45, 50] as examples. Predictor-corrector methods that alter a trained generative model in the decoding process of a diffusion model have also been studied (see [65]). Non diffusion model based methods have also been studied (see "Markov Chain Generative Adversarial Neural Networks for Solving Bayesian Inverse Problems in Physics Applications").
>
> An advantage of our paper is its independence from the choice of posterior sampling algorithm. Namely, our results do not depend on the algorithm used, and analysis of the sampling algorithm can be done separately, for instance in "Diffusion Posterior Sampling for Linear Inverse Problem Solving". We believe that further studies including analysis of these algorithms are interesting and important problems. In the future, we aim to examine these implementations for precise verifications of our theory and generalizations of our results.
>
> **We will add this discussion in §5 ("Limitations and future work").**
>
> # Questions:
>
> ## When the prior is...
>
> This is an excellent question. You are correct in your understanding. **We will add the following after Thm 3.5:**
>
> "Suppose that $\mathcal{P} = \mathcal{R}$ is a sum of $d$ Diracs located at $X = \{x_1,\ldots,x_d\} \subseteq \mathbb{R}^n$. Then Theorem 3.5 predicts recovery from roughly $m = O(\log(d))$ measurements. Intuition for this comes from the Johnson-Lindenstrauss Lemma (JLL). The matrix $A \in \mathbb{R}^{m \times n}$ is a linear dimensionality-reducing map. JLL states that distances in $X$ are preserved under $A$ if and only if $m = O(\log(d))$. In this setting, preserving the distances in $X$ is equivalent to being able to stably identify the mode from which a signal is drawn."
>
> ## Line 328; is there any...
>
> Yes, good point. Theoretically, according to Thm 3.8, the smaller the coherence the fewer measurements are required. Empirically, [14] conducts numerical experiments in the deterministic case that show that incoherent generative models require fewer measurements for recovery. **We will add the following clarification at the end of l.328:**
>
> "Numerically, they show that generative models with smaller coherence achieve better recovery from the same number of measurements than those with larger coherence."
>
> In future work, we also intend to study this phenomenon experimentally in the Bayesian setting.
>
> ## It is worth reflecting on...
>
> Thanks for this insightful comment and references. This is very interesting work, and there are definite overlaps with ours. In a general, both approaches strive to consider arbitrary prior distributions and incorporate distributional information into the recovery. However, there are also some key differences. First, our results are nonasymptotic - they hold for fixed $m$ and $n$ - whereas most of the work on AMP considers an asymptotic regime. Second, much of the AMP literature focuses on Gaussian measurement matrices. Our focus is more on non-Gaussian matrices, e.g. subsampled orthogonal matrices. We are not aware of as complete a theory for AMP-type techniques for such matrices. Third, in our work we consider Bayesian recovery, where the aim is to approximate the posterior. As far as we are aware, AMP methods typically compute a point estimator for $x^*$ only. However, it's worth noting that AMP methods are fast algorithms for computing point estimators, whereas we do not consider algorithms for sampling from the posterior. As we commented above, efficient implementation of posterior sampling is an active area of investigation.
>
> **We will add this discussion and references to §1.4 ("Related work").**
>
> ## For example, in the asymptotic....
>
> Excellent question. This was studied in [46], which gives information-theoretic lower bounds. [46] shows that if any recovery scheme achieves $O(\eta)$ accuracy then the number of measurements (for any $A$) must scale at least linearly in $\log \mathrm{Cov}_{\eta,\delta}$. Combined with the upper bound for Gaussian matrices, the conclusion is that the approximate covering number sharply characterizes the number of measurements for recovery (up to constants).
>
> The focus of our paper is upper bounds, and, as noted, non-Gaussian measurement matrices. Nonetheless, it is important to note that such lower bounds exist. **We will add a comment on this to §1.4 as well.**
>
> ## As a matter of clarity, it...
>
> Excellent comment. You are correct that $m$ enters the concentration bounds, which are exponentially-decaying in $m$ for reasonable $\mathcal{A}$, $\mathcal{E}$.  **We will add the following remark about this immediately after Thm 3.1:**
>
> "Since Thm 3.1 considers arbitrary measurement and noise distributions, the number of measurements $m$ does not explicitly enter the bound. For typical distributions, a dependence on $m$ is found in the concentration bounds $C_{\mathsf{low}},C_{\mathsf{upp}},D_{\mathsf{upp}}$, as well as the terms $D_{\mathsf{shift}}$ and $C_{\mathsf{abs}}$. In particular, the former decay exponentially-fast in $m$ for the examples introduced in §1.2 and therefore compensate for the exponentially-large scaling in $k$ in the main bound. See §B for precise estimates. See also the discussion in §3.1 and 3.2 below."
>
> ## It would also be....
>
> Good point. As there may not be space in the main paper, **we will add a clear description at the beginning of §C**.
>
> Allow us to briefly explain. The first step in the proof of Thm 3.1 is to decompose $\mathcal{R},\mathcal{P}$ into distributions $\mathcal{R}',\mathcal{P}'$ that are supported in balls of a given radius, plus remainder terms. The distributions $\mathcal{R}',\mathcal{P}'$ are close (in $W_{\infty}$) to a distribution $\mathcal{Q}$ supported on the centres of the balls that give the approximate cover satisfying eqn (3.1).
>
> The next step is to replace $x^* \sim \mathcal{R}$ in the definition of $p$ in Thm 3.1 by $z^* \sim \mathcal{P}'$. The is done to align the prior with the posterior $\mathcal{P}(\cdot | y , A)$, which is needed later in the proof. We do this using Lemma C.6. Here, we have to consider the action of $A$ on vectors $x - z$, where $x \in \mathrm{supp}(\mathcal{R}')$ and $z \in \mathrm{supp}(\mathcal{P}')$. After determining the supports of $\mathcal{R}',\mathcal{P}'$, we see that $x - z \in D_1$, where $D_1$ is the set in eqn (3.2).
>
> The set $D_2$ in eqn (3.2) comes later in the proof. We decompose $\mathcal{P}'$ into a mixture over the balls mentioned above. After a series of steps, we reduce the problem to considering the probability that the conditional distribution is drawn from one ball when the prior is drawn from another. Lemmas C.4-C.5 handle this. They involve estimating the action of $A$ on vectors $z - x$, where $z$ is the centre of one of the balls and $x$ comes from another ball. Since the balls are supported in $\mathrm{supp}(\mathcal{P})$ we have $x \in \mathrm{supp}(\mathcal{P})$ and since the come from the approximate covering number bound (3.1), $z \in \mathrm{supp}(\mathcal{P})$ if $\mathcal{P}$ attains the minimum and $z \in \mathrm{supp}(\mathcal{R})$ otherwise. Hence $z - x \in D_2$, with $D_2$ as in (3.3).
>
> ## Wherever you have...
>
> Excellent question. The short answer is no: we did mean to write $\mathrm{supp}(\mathcal{P}) - \mathrm{supp}(\mathcal{P})$. The key point is that if $\mathcal{P}$ attains the minimum in (3.1) then the set $D_2$ in (3.3) depends on $\mathcal{P}$ only.
>
> Recall that $\mathcal{R}$ is the 'real' distribution, which is typically unknown, whereas $\mathcal{P}$ is the 'approximate' distribution and is known. It is convenient to have results in terms of $\mathcal{P}$, as this means the terms appearing in the recovery guarantees can be estimated numerically or theoretically. We discuss this further below.
>
> As an aside, note that whether $\mathrm{supp}(\mathcal{P}) - \mathrm{supp}(\mathcal{R})$ or $\mathrm{supp}(\mathcal{P}) - \mathrm{supp}(\mathcal{P})$ arises is determined by which set $\mathcal{R}$ or $\mathcal{P}$ has smaller covering number. See eqn (3.1). This arises because it is the smallest set over which our concentration bounds are formulated. As we commented in Remark 3.2, one cannot generally expect good concentration bounds over the whole of $\mathbb{R}^n$. Therefore, it is important to formulate them over as small sets as possible.
>
> We believe it is useful to have results in terms of $\mathcal{P}$ only. First, if $\mathcal{P}$ is a generative model, we can estimate its approximate covering number analytically. See Proposition 4.1. Second, in Thm 3.8 the coherence in (3.5) is taken over $D = \mathrm{supp}(\mathcal{P}) - \mathrm{supp}(\mathcal{P})$. This has several benefits. First, one can derive analytical bounds in some cases. For example, [14] does this for ReLU generative models with random weights and biases. Second, one can estimate it empirically by, e.g., drawing a large sample from $\mathcal{P}$ and replacing the supremum in Defn 3.7 by a maximum. This allows one to gauge empirically how well one can recover with a given $\mathcal{P}$. Note that this may not be possible if one considered $\mathcal{R}$, as we may not be able to sample from it.
>
> We agree this point should be better highlighted. **We will add a comment about this after Thm 3.8.**

---

> > ### Comment · Reviewer_b54M · 2025-08-03
> >
> > I appreciate the clarifying remarks provided by the authors. I will raise my score to a 5.

---

> > > ### Author Response · Authors · 2025-08-04
> > > **Thanks!**
> > >
> > > We very much appreciate your careful review and reading of our rebuttal and are delighted that you will raise your score. Thanks!

---

### Official Review · Reviewer_aqBr · 2025-07-16

**Clarity:** 3
**Significance:** 3
**Originality:** 3
**Rating:** 5
**Confidence:** 4

**Summary:**

The authors establish rigorous recovery guarantees for compressed sensing when signals are drawn from a known prior and both the measurement operator and noise follow arbitrary distributions. Their main result (Theorem 1) bounds the probability that the reconstruction error of a posterior sample exceeds a fixed constant plus a noise-dependent term. They then instantiate this general bound to obtain explicit recovery guarantees for two important forward operators subgaussian matrices and subsampled orthonormal transforms which respectively extend the results of [3,14] and [46]. Finally, they show that the same framework applies to two practically relevant approximate priors: Lipschitz-continuous generative models and sparse vectors.

**Questions:**

1. **Could the authors please outline their technical contributions that enabled them to generalize the arguments in \[3, 14, 46]?**

2. **Could the authors explain the technical difficulty of assuming a distribution over the noise term instead of specifying a density, as mentioned in the “Future Work” section?**

**Ethical Concerns:**

["NO or VERY MINOR ethics concerns only"]

**Final Justification:**

The authors have alleviated my concerns and demonstrated the technical strength to make this paper a strong contribution to the community. It generalizes the theoretical arguments of several influential works on generative priors for inverse problems and would be an important paper for the research community.

**Quality:**

3

**Strengths And Weaknesses:**

**Strengths**

1. The work is dense, but easy to follow because it is well written and tells a coherent story.
2. It generalizes impactful results from previous work \[3,14,46], as shown in Section 3.
3. It provides explicit sample complexity bounds for Lipschitz generative neural network priors.

**Weaknesses**

1. The main weakness is that the authors do not explain any novel proof techniques they used to develop these generalization methods; the manuscript’s tone suggests they are simply leveraging arguments from prior work.
2. There is no remark, justification, or explanation for why the noise distribution is left unrestricted in Definition 2.5.

---

> ### Author Rebuttal · Authors · 2025-07-29
>
> We very much appreciate your effort in reviewing our paper and are delighted by your positive assessment. Here we address the comments in your report.
>
> # Weaknesses
>
> ## 1. The main weakness is that the authors do not explain any novel proof techniques they used to develop these generalization methods; the manuscript’s tone suggests they are simply leveraging arguments from prior work.
>
> Thanks for the comment. We discuss this further below in our reply to your question.
>
>
>
>
> ## 2. There is no remark, justification, or explanation for why the noise distribution is left unrestricted in Definition 2.5.
>
> Thanks for this comment. Arguably, having a general noise distribution is not the biggest innovation in the paper, as in many applications the noise is assumed to be standard Gaussian. We agree the results are `cleaner' with Gaussian noise, and in fact we used this in our simplified results, Theorems 1.1-1.3. However, since a major focus of our work was to allow for arbitrary distributions $\mathcal{R},\mathcal{P},\mathcal{A}$ it seemed strange to then consider only standard Gaussian noise in the main theorem. Unfortunately, as can be seen in Definition 2.5, we could not consider completely arbitrary noise - however, see also the discussion below. Regardless, we do think there are some benefits to doing this. For example, if the noise $\mathcal{E} = \mathcal{N}(0,\Sigma)$ was Gaussian but with a a nontrivial covariance matrix (as arises in some applications), then we can still apply our results. Conversely, if one had only considered standard Gaussian noise, the proofs would have had to be redone for each new noise distribution. We agree a better justification could be added. Due to space limitations, we did not do this in the paper. **We will add a short justification/explanation to the revision, immediately after Theorem 3.1.**
>
> # Questions:
>
> ## 1. Could the authors please outline their technical contributions that enabled them to generalize the arguments in [3, 14, 46]?
>
> Thanks for this question. You are correct that our work is most closely related to [3,14,46].
>
> The works [3,14] consider the deterministic setting, where $x^*$ does not follow a distribution. They derive recovery guarantees for the point estimator $\hat{x}$ obtained by minimizing the $\ell^2$-loss. For a more detailed discussion on this setting, please see l.143-148. Note that this discussion is in reference to [20], but the same considerations apply to [3,14].
>
> We want to stress that the proof techniques employed in the Bayesian case are completely different to those used in the deterministic case [3,14]. Unfortunately, at several points in the paper we referred to our work as 'generalizing' [3,14]. This may have given the wrong impression. Our work derives a Bayesian 'analogue' of [3,14] rather than a `generalization' per se. This aside, both [3,14] also make assumptions on the model class $\Sigma$ (analogous to $\mathrm{supp}(\mathcal{P})$ in our work). In [14] $\Sigma$ is the range of a ReLU generative model. [3] is more general, but it still assumes that $\Sigma$ a subset of finite union of finite-dimensional subspaces. These works then use matrix concentration inequalities and union bounds [14] or chaining arguments [3] to derive recovery guarantees. We do not consider these specific models, or distributions related to them, nor do we use these techniques. The main concept we use from [3,14] is the coherence (Definition 3.7), which appears in Theorem 3.8. Our work is also inspired by [3] in the sense that it strives to be very general (in fact, more general than [3] -- see above), by considering arbitrary distributions $\mathcal{P}$. However, we reiterate that there no overlap with the proof techniques employed therein.
>
>
> Our work is a generalization of [46], which considered the case where $\mathcal{A}$, $\mathcal{E}$ were Gaussian. Our proof techniques are certainly built on [46], however they involve significant modifications and nontrivial extensions to handle general $\mathcal{A},\mathcal{E}$. Perhaps the biggest technical contribution we make is encompassed by the introduction of the concentration bounds, Definitions 2.3 and 2.4. The key innovations here are threefold. First, we show that these concentration bounds, for general $\mathcal{A},\mathcal{E}$, precisely control the quality of recovery -- see the main bound in Theorem 3.1. Second, we impose the concentration bounds for $\mathcal{A}$ over subsets of $\mathbb{R}^n$ only. Please see $D_1,D_2$ in Theorem 3.1 and Remark 3.2 for more discussion. Here we would like to stress that it is not necessary to do this in the Gaussian case considered in [46], as one can get fast concentration over the whole of $\mathbb{R}^n$ with a Gaussian random matrix. But it is absolutely crucial for other distributions, e.g. subsampled orthogonal matrices. Please see Remark 3.9. Showing that concentration was only required over sets related to the support of $\mathcal{P}$ (see $D_1$ and $D_2$ in Theorem 3.1) was a major technical hurdle to overcome in this work. Had we only considered concentration over $\mathbb{R}^n$, then the  measurement condition in the subsampled orthogonal matrix case would have been at best $m \gtrsim n \log(n)$ -- a largely meaningless bound. Third, it was necessary to introduce both relative $C_{\mathsf{low}}$, $C_{\mathsf{upp}}$ concentration bounds and, to our surprise, an absolute $C_{\mathsf{abs}}$ concentration bound as well. Had we only considered the former two, we would have again ended up with meaningless bounds.
>
>
> We agree these points could have been made clearer in the paper. **We will amend and expand the discussion in §1.4 on [3,14,46], emphasizing that our work does not generalize [3,14] and providing some more details (as discussed above) on the technical contributions in relation to [46].**
>
>
>
>
>
>
> ## 2. Could the authors explain the technical difficulty of assuming a distribution over the noise term instead of specifying a density, as mentioned in the “Future Work” section?
>
> Thanks for this question. The reason for specifying a density for the noise comes from a key step, Lemma C.6, in the proof of Theorem 3.1 where we need to replace the `real' distribution $\mathcal{R}$ in the probability term $p$ in Theorem 3.1 by the approximate distribution $\mathcal{P}$. This is done in order to align the prior and the posterior, which is necessary for the subsequent steps of the proof. The density shift bound arises in the proof of Lemma C.6 when we make a change of variables. Specifically, given $e \sim \mathcal{E}$ and $( x^* ,z^* ) \sim \Pi$, where $\Pi$ is a coupling of $\mathcal{R}$ and $\mathcal{P}$, we introduce a new variable $e' = e + A(x^* - z^*)$. To handle this change of variables, we assume that the noise has a density and use the density shift bound $D_{\mathsf{shift}}$ to bound the effect of this change.
>
> We agree this term is a bit mysterious, and are looking into whether it can be removed in the future. **We will add some more details about where this assumption comes from in §5 (``Limitations and future work'').**

---

### Decision · Program_Chairs · 2025-09-17

**Decision:**

Accept (spotlight)

**Comment:**

The authors establish recovery guarantees for linear inverse problems with prescribed prior for the unknown when both the measurement operator and noise follow arbitrary distributions.  They specialize their approach to obtain guarantees in cases where forward operators are subgaussian matrices and subsampled orthonormal transforms, and finally tackle particular cases of Lipschitz-continuous generative models and sparse vectors. The paper was referred by four reviewers who all recommended acceptance, and I do so too.